# Local Minima in Quadratic-Penalty Relaxations of Binary Linear Programs

**Cheng-Han Huang** [1]  **Yongliang Sun** [1]  **Chaoyan Huang** [1 2]  **Ismail Alkhouri** [3 4]  **Rongrong Wang** [1 5]

## Abstract

Many combinatorial optimization problems admit quadratic unconstrained binary formulations (QUBO) which can often be relaxed to the box $[0, 1]^n$ and optimized using scalable gradient-based methods. However, the resulting non-convex landscape can often contain local optima that are spurious or infeasible. In this paper, we establish sufficient structural conditions on quadratic penalties that rule out these failures, guaranteeing that every local minimizer of the relaxed problem is both binary and feasible. For each problem we study, we examine existing QUBO formulations when available, identify why they fail when they do, and propose alternative relaxed QUBOs that satisfy our conditions. We show for several common combinatorial problems, including open-pit mining, 0–1 knapsack, and traveling salesman formulations, that these constructions allow gradient-based methods such as projected gradient descent and Adam to be safely applied to obtain valid binary solutions. Our results clarify when differentiable optimization is a reliable local solver for quadratic combinatorial objectives.

## 1. Introduction

Quadratic unconstrained binary optimization (QUBO) formulations provide a unifying framework for a wide range of combinatorial optimization problems (Kochenberger et al., 2014), including maximum independent set (Karp, 2009), knapsack variants (Martello & Toth, 1990), open-pit mining

[1]Department of Computational Mathematics, Science, & Engineering, Michigan State University [2]Department of Electrical Engineering and Computer Science, University of Michigan [3]XCP, Los Alamos National Laboratory [4]Michigan Institute for Computational Discovery and Engineering, University of Michigan [5]Department of Mathematics, Michigan State University. Correspondence to: Cheng-Han Huang <huang248@msu.edu>, Rongrong Wang <wangron6@msu.edu>.

*Proceedings of the $43^{rd}$ International Conference on Machine Learning*, Seoul, South Korea. PMLR 306, 2026. Copyright 2026 by the author(s).

(Lerchs, 1965), and the traveling salesman problem (Dantzig et al., 1954). Traditionally, QUBOs have been solved using discrete optimization techniques such as local bit-flip heuristics (Benlic & Hao, 2013), simulated annealing (Kirkpatrick et al., 1983), branch-and-bound methods (Clausen, 1999), or quantum-inspired algorithms (Glover et al., 2022b). In these settings, optimization is performed directly over binary variables, and feasibility or optimality is determined purely by objective values on discrete configurations. For a QUBO encoding of a constrained combinatorial problem, feasibility refers to satisfaction of the original discrete constraints encoded by the formulation.

Recently, however, there has been growing interest in optimizing relaxed QUBOs over the continuous domain $[0, 1]^n$ using scalable gradient-based methods (Alkhouri et al., 2025; 2026; Sun et al., 2026). This approach is attractive due to its compatibility with modern hardware (GPU/TPU) and automatic differentiation frameworks (Baydin et al., 2018). Nevertheless, a fundamental obstacle remains: local minimizers of relaxed QUBOs are not guaranteed to correspond to valid binary and feasible solutions of the original combinatorial problem. Even when constraint violations are penalized with arbitrarily large weights, relaxed QUBOs may admit fractional or infeasible *strict local minima* under box constraints. This phenomenon cannot be explained by insufficient optimization or poor initialization; rather, it reflects intrinsic geometric properties of the nonconvex penalty landscape. As a result, continuous optimization of relaxed QUBOs is often regarded as unreliable for exact combinatorial solving.

**Central question.** This paper addresses the following question: *When can gradient-based optimization of a relaxed QUBO be trusted as a local solver for the original discrete problem?* More precisely, we seek structural conditions under which every local minimizer of a box-relaxed QUBO is both binary and feasible.

Importantly, we do not require global optimality, convexity, or exact penalty equivalence. Our focus is exclusively on *local minima* under box constraints, which govern the behavior of projected gradient descent (PGD) and related first-order methods.

**Why large penalties are insufficient.** A common intuition is that infeasible solutions can always be eliminated

by increasing the penalty parameter (Nocedal, 2006). We show that this intuition is false in the continuous relaxation. Specifically, we identify a failure mode in which the penalty landscape becomes "trapped" at the boundary because the gradient points "outward" from the feasible region, rather than toward it. Such a point becomes a strict local minimizer regardless of how large the penalty weight is.

This phenomenon arises in standard QUBO formulations, including widely used penalties for precedence constraints in open-pit mining. It demonstrates that feasibility at local minima is not determined by penalty magnitude alone, but by the *first-order geometry* of the penalty.

**Our approach.** We show that the correctness of local minima in relaxed QUBOs can be guaranteed by simple, verifiable structural properties of the penalty. Our analysis identifies three key ingredients:

- diagonal-free quadratic penalties on core variables,

- integer-valued penalty gradients at binary points, and

- a local repairability condition ensuring that every infeasible binary point admits a feasible descent direction for the penalty with a uniform margin.

Under these conditions, we prove that for sufficiently large (but finite) penalty weight, every local minimizer of the relaxed problem is both binary and feasible. These guarantees are purely local, require no convexity assumptions, and apply directly to projected gradient methods.

**Implications.** Our framework explains both successes and failures of existing QUBO formulations. It recovers known guarantees for maximum independent set as a special case (such as those in (Mahdavi Pajouh et al., 2013; Alkhouri et al., 2025)), explains why common formulations for open-pit mining can fail even with infinite penalty, and motivates new penalty constructions for knapsack and traveling salesman problems that admit feasible local minima. Rather than proposing problem-specific heuristics, our results clarify which geometric properties any QUBO penalty must satisfy to be compatible with gradient-based optimization.

This paper makes the following contributions:

- **Structural obstruction.** We identify a fundamental failure mode in relaxed QUBOs: infeasible binary points that are first-order stationary under box constraints and therefore remain strict local minima for arbitrarily large penalty weights. This shows that increasing the penalty magnitude alone is insufficient to guarantee feasibility under continuous optimization.
- **General local correctness theory.** We establish problem-independent conditions under which every

local minimizer of a relaxed QUBO is binary and feasible. Our results rely on diagonal-free quadratic penalties with integer coefficients and a local repairability property that rules out infeasible stationary points.

- **Binarity and feasibility guarantees.** We prove separate theorems showing that sufficiently large penalties eliminate fractional local minima and infeasible binary local minima, yielding finite, explicit penalty thresholds.

- **Explanatory power for existing formulations.** Our framework recovers known guarantees for maximum independent set, explains the failure of commonly used penalties for open-pit mining, and clarifies when standard QUBO constructions are incompatible with gradient-based methods.

- **New penalty constructions.** Guided by our theory, we propose alternative QUBO penalties for knapsack and traveling salesman formulations that satisfy the required structural conditions and admit only valid local minima under continuous relaxation.

## 1.1. Related studies

Classical QUBO formulations prioritize binary-domain correctness, as they are tailored for discrete solvers (e.g., branch-and-bound, quantum heuristics) (Glover et al., 2022a). However, they often fail under gradient-based continuous relaxation because they do not account for the first-order geometry of the box $[0, 1]^n$. This work analyzes local minima in relaxed QUBOs and identifies structural penalty properties that ensure local correctness. By bridging discrete formulations with continuous optimization, we provide a theoretical foundation for using gradient methods as reliable combinatorial solvers.

The paper in (Burer & Letchford, 2009) studies nonconvex quadratic programming problems with box constraints and shows that finding a global optimum is NP-hard. It also provides a detailed characterization of the extreme points of related linearized formulations. While both (Burer & Letchford, 2009) and our work analyze properties of box-constrained quadratic objectives, our paper focuses on their behavior under projected gradient-based optimization methods.

In recent studies (Alkhouri et al., 2025; 2026), the authors proposed using differentiable solvers for relaxed QUBOs applied to the Maximum Independent Set (MIS) and Maximum Cut (MaxCut) problems. While these works characterize binary local optima, including stationary points, their analyses are inherently problem-specific. In contrast, our paper generalizes the analysis of local optima for relaxed QUBOs by deriving problem-agnostic properties and, in addition, proposing new theory-guided formulations.

# 2. Main Theory: Local Correctness of Relaxed QUBOs

## 2.1. Problem Setting

We consider relaxed QUBO objectives of the form

$$\hat{f}(z) = w^\top z + \gamma V(z), \quad z \in [0,1]^N, \quad N := n+m, \quad (1)$$

where $z = (x,y)$ collects decision variables $x \in \mathbb{R}^n$ (i.e., the variables that appear in the linear objective term) and possible auxiliary variables $y \in \mathbb{R}^m$, $w$ is an expanded linear objective on the variables, with weight $0$ assigned to all auxiliary variables, and $V(z)$ is a quadratic penalty encoding feasibility constraints. If no auxiliary variables are involved, we simply drop the $y$ term and refer to $z = x$. The parameter $\gamma$ controls the relative strength of the penalty landscape. In particular, variables with zero objective weight do not affect the original objective value. We therefore refer to the decision variables with nonzero objective weight as the *core variables*.

Our goal is to characterize the structural conditions in $V$ in which, provided $\gamma$ is sufficiently large, *every local minimum of the PGD of $\hat{f}$ over $[0,1]^N$ corresponds to a binary and feasible solution.*

Let's first review some preliminaries, which will be used in our model design and analysis.

**Definition 2.1** (Projected Gradient Descent (PGD)). Given an initial point $z^0 \in [0,1]^N$ and a step size sequence $\{\eta_k\}_{k \geq 0}$ with $\eta_k > 0$, PGD for minimizing $\hat{f}$ over $[0,1]^N$ generates iterates

$$z^{k+1} = \Pi_{[0,1]^N}\left(z^k - \eta_k \nabla \hat{f}(z^k)\right), \quad (2)$$

where $\Pi_{[0,1]^N}$ denotes the Euclidean projection onto the box $[0,1]^N$, applied componentwise.

**Definition 2.2** (Box-stationary point of PGD). A point $z^\star \in [0,1]^N$ is called a *box-stationary point of PGD* for a continuously differentiable function $f$ if it satisfies the following first-order condition

$$\langle \nabla f(z^\star), d \rangle \geq 0, \quad \forall d \in T(z^\star),$$

where $T(z) := \{d \in \mathbb{R}^N : z_i = 0 \Rightarrow d_i \geq 0, \ z_i = 1 \Rightarrow d_i \leq 0, \ 0 < z_i < 1 \Rightarrow d_i \in \mathbb{R}\}$, which means $T(z^\star)$ denotes the tangent cone of the box $[0,1]^N$ at $z^\star$.

When a box-stationary point further satisfies the second order sufficient condition, it is a *box-local minimizer*. A box-local minimizer admits no feasible first-order descent direction under the box constraint, and therefore any PGD limit point that is a box-local minimizer is a valid discrete solution.

**Definition 2.3** (Penalty interaction graph). Let

$$V(z) = \sum_{1 \leq i \leq j \leq N} Q_{ij} z_i z_j + \sum_{i=1}^{N} d_i z_i + \text{const},$$

be a quadratic penalty. The *penalty interaction graph* is the undirected graph $G = (\{1, \ldots, N\}, E)$, where $(i,j) \in E$ if and only if $Q_{ij} \neq 0$.

## 2.2. Interior Independence of Local Minima

We first establish a useful lemma showing that under core diagonal-free penalties (i.e., quadratic penalties with no $z_i^2$ terms on the core variables), the set of interior coordinates at a PGD local minimum forms an independent set in the penalty interaction graph.

**Lemma 2.4** (Interior independence). *Let $\hat{f}(z) = w^\top z + \gamma V(z)$ with $V$ diagonal-free on the core variables. Let $\mathcal{C} := \{i : w_i \neq 0\}$ be the set of nonzero-weight (core) indices. If $z^\star$ is a local minimizer of $\hat{f}$ over $[0,1]^N$, $i \in \mathcal{C}$ is any interior core coordinate, and $Q_{ij} \neq 0$, then $z_j^\star \in \{0,1\}$. In particular, core interior coordinates form an independent set in the interaction graph of $V$.*

## 2.3. Binarity of Local Minimizers

We are now ready to show that under diagonal free penalties, a sufficiently large penalty coefficient eliminates fractional local minimizers on all variables with nonzero linear weight.

**Theorem 2.5** (Core binarity at local minimizers). *Let*

$$\hat{f}(z) = w^\top z + \gamma V(z), \quad z \in [0,1]^N,$$

*where*

$$V(z) = \sum_{1 \leq i \leq j \leq N} Q_{ij} z_i z_j + \sum_{i=1}^{N} d_i z_i + \text{const}$$

*is quadratic, diagonal-free on the core variables, and has integer coefficients $Q_{ij}, d_i \in \mathbb{Z}$. Let $\mathcal{C} := \{i : w_i \neq 0\}$ be the set of core indices. Assume*

$$|\gamma| > \max_{i \in \mathcal{C}} |w_i|. \quad (3)$$

*Then every local minimizer $z^\star$ of $\hat{f}$ over $[0,1]^N$ satisfies*

$$z_i^\star \in \{0,1\} \quad \text{for all } i \in \mathcal{C}.$$

## 2.4. Feasibility via Local Repairability

Binarity alone does not guarantee feasibility. We now show that feasibility of local minima can be enforced by a purely local geometric condition on the penalty.

**Definition 2.6** (Local repairability with margin $\delta > 0$). Let $\mathcal{F} \subseteq \{0,1\}^{|\mathcal{C}|}$ denote the feasible set of core assignments, and let $\mathcal{D} \subseteq \mathbb{R}^N$ be a bounded family of candidate directions. We say that $V$ is *locally repairable with margin $\delta > 0$* (relative to $\mathcal{D}$) if for every

$$z = (x, y) \in [0,1]^N$$

whose core satisfies

$$x \in \{0,1\}^{|\mathcal{C}|} \setminus \mathcal{F},$$

there exists a direction $d \in \mathcal{D} \cap T(z)$ such that

$$\langle \nabla V(z), d \rangle \leq -\delta.$$

If there are no auxiliary variables, simply read $z = x$.

**Theorem 2.7** (Feasibility of binary local minimizers). *Consider*

$$\hat{f}(z) = w^\top z + \gamma V(z), \qquad z \in [0,1]^N,$$

*and assume:*

  (i) (**Core binarity**) *Every local minimizer of $\hat{f}$ over $[0,1]^N$ has a binary core.*

  (ii) (**Local repairability**) *$V$ is locally repairable with margin $\delta > 0$ relative to a bounded direction set $\mathcal{D}$.*

*Define*

$$L := \sup \Big\{ \langle w, d \rangle : z \in [0,1]^N \text{ has infeasible binary core,}$$
$$d \in \mathcal{D} \cap T(z) \Big\}.$$

*Then, for every*

$$\gamma > \frac{L}{\delta},$$

*every local minimizer of $\hat{f}$ over $[0,1]^N$ has feasible core.*

### 2.5. A Unified Criterion for Local Correctness

We now summarize the preceding results into a unified criterion that guarantees the local correctness of relaxed QUBOs under projected gradient descent.

Consider the box-constrained optimization problem of (1), we have

$$\min_{z \in [0,1]^N} \hat{f}(z) := w^\top z + \gamma V(z).$$

Our analysis identifies three structural conditions on the penalty $V$ that together ensure local correctness.

**(C1) Diagonal-free structure.** The quadratic penalty $V$ is diagonal-free on the core variables, i.e., it contains no $z_i^2$ terms for core indices. This structural property implies an *interior independence* phenomenon: at any PGD local minimum, no two interacting variables can simultaneously lie in the interior of the box.

**(C2) Integer-valued penalty gradients.** The coefficients of $V$ are integer-valued. Combined with interior independence, this ensures that the penalty gradient at any interior core coordinate is integer-valued. For sufficiently large penalty weight $\gamma$, this eliminates fractional local minima on all core variables with nonzero linear weight, yielding core binarity.

**(C3) Local repairability.** The penalty $V$ is locally repairable with margin $\delta > 0$ relative to a bounded family of admissible directions. That is, from any infeasible binary core configuration, there exists a feasible first-order descent direction that strictly decreases the penalty.

Under these three conditions, the geometry of the relaxed objective excludes both fractional and infeasible PGD local minima. Hence, we have the following main theorem.

**Theorem 2.8** (Unified criterion for local correctness). *Assume that the quadratic penalty $V$ satisfies* (C1)–(C3) *above. Then there exists a finite threshold $\gamma^\star > 0$ such that for all $\gamma > \gamma^\star$, every box-local minimizer of $\hat{f}$ over $[0,1]^N$ has a binary core belonging to the projected feasible set of the original combinatorial problem.*

Thus, under the stated assumptions, invalid box-local minima are ruled out. This criterion is purely local, requires no convexity assumptions, and is independent of the specific combinatorial problem under consideration.

## 3. Examples

We apply our theorem to several classical combinatorial optimization (CO) problems with linear objectives and hard constraints, illustrating how our penalty construction ensures binarity and feasibility of local minimizers. We always cast examples into the minimization template $\hat{f}(z) = w^\top z + \gamma V(z)$.

### 3.1. Open-Pit Mining (OP)

Open-pit mining is a classical precedence-constrained combinatorial optimization problem arising in large-scale resource extraction (Hochbaum & Chen, 2000). The mining region is discretized into a finite collection of blocks, each associated with an economic value. Due to slope stability and safety requirements, the extraction of a given block is permitted only after certain predecessor blocks have been removed. The objective is to select a subset of blocks that maximizes the total economic value while respecting all

precedence constraints.

Let $i \in \{1, \ldots, n\}$ index the blocks. Each block $i$ is associated with an economic value $w_i \in \mathbb{R}$, and a binary decision variable $x_i \in \{0, 1\}$ indicating whether the block is extracted.

For each block $i$, let $P(i) \subseteq \{1, \ldots, n\}$ denote the set of predecessor blocks that must be extracted before $i$ (e.g., due to slope or stability constraints). The precedence-constrained optimization problem is given by

$$\max_{x \in \{0,1\}^n} \quad \sum_{i=1}^n w_i x_i,$$

$$\text{s.t.} \quad x_i \leq x_j, \qquad \forall i \in \{1, \ldots, n\}, \ \forall j \in P(i).$$

We begin with a naive QUBO formulation for open-pit mining that enforces precedence constraints only between parent–child pairs. This formulation is often used as a direct encoding of the constraints, and we therefore refer to it as the *naive parent formulation*.

**Naive parent formulation.** Let $p \to c$ denote a precedence constraint requiring that block $p$ be extracted before block $c$. The naive parent formulation uses the penalty

$$V_{\mathrm{par}}(x) := \sum_{p \to c} x_c(1 - x_p),$$

leading to the relaxed objective

$$\hat{f}(x) = -w^\top x + \gamma V_{\mathrm{par}}(x).$$

As we will show, this seemingly natural penalty can admit infeasible strict local minima in the box relaxation, regardless of how large the penalty parameter $\gamma$ is. This demonstrates that feasibility does not automatically follow from large penalty weights unless the penalty satisfies an explicit local repairability condition.

We construct an alternative QUBO penalty that is specifically designed to satisfy the sufficient conditions for local correctness.

**Ancestor formulation (theory-guided).** Instead of penalizing only immediate parent–child violations, we require that whenever a block $c$ is extracted, all of its ancestors must also be extracted. Let $Anc(c)$ denote the set of all (strict) ancestors of node $c$ in the precedence directed acyclic graph (DAG). We define the penalty

$$V_{\mathrm{anc}}(x) := \sum_{c \in \{1,2,\ldots,n\}} \sum_{\alpha \in Anc(c)} x_c(1 - x_\alpha),$$

and the corresponding relaxed objective

$$\hat{f}(x) = -w^\top x + \gamma V_{\mathrm{anc}}(x). \tag{4}$$

We can show that the ancestor formulation allows each infeasible point to admit one-bit repairable directions and is therefore compatible with the local repairability criterion developed earlier.

For both QUBO formulations, the binarity condition in our framework is satisfied, as the quadratic penalties are diagonal-free and all coefficients of $V$ are integer-valued.

**Proposition 3.1** (Feasibility of the ancestor formulation under a finite penalty). *Let $\hat{f}(x) = -w^\top x + \gamma V_{\mathrm{anc}}(x)$ over $[0,1]^n$. Assume that core binarity holds. Then $V_{\mathrm{anc}}$ is locally repairable with margin $\delta = 1$ relative to the single-bit removal directions*

$$D := \{-e_i : i = 1, \ldots, n\}.$$

*Then any $\gamma > \max_i w_i$ guarantees that every binary PGD local minimum of $\hat{f}$ has a feasible core.*

By Thm 2.8, any penalty parameter $\gamma$ satisfying both the core binarity condition of Thm 2.5 and the feasibility condition of Thm 2.7 guarantees that all PGD local minimizers are binary and feasible. For the open-pit mining (4), this reduces to the simple sufficient condition

$$\gamma > \max_i |w_i|.$$

For the ancestor formulation, turning off a violating node always reduces the penalty. Flipping $x_c : 1 \to 0$ eliminates all terms $x_c(1 - x_\alpha)$ for every missing ancestor $\alpha$, and the penalty drops by at least $1$. We now contrast this positive result with a failure mode of the naive parent formulation.

*Remark* 3.2 (Why the naive parent formulation can fail). Consider the chain $g \to r \to a \to b$ and the infeasible binary point

$$x_g = 0, \quad x_r = 0, \quad x_a = 1, \quad x_b = 1.$$

At this point, $V_{\mathrm{par}}(x) = x_r(1 - x_g) + x_a(1 - x_r) + x_b(1 - x_a) = 1$, yet a direct calculation shows that

$$\nabla V_{\mathrm{par}}(x) = 0.$$

Hence, for every feasible direction $d \in T(x)$ we have $\langle \nabla V_{\mathrm{par}}(x), d \rangle = 0$, violating the local repairability condition for any $\delta > 0$. This flatness reflects the need for coordinated multi-bit repairs in the parent formulation, in contrast to the strict one-bit repairs admitted by the ancestor formulation.

**Lemma 3.3** (The flat infeasible vertex can be a strict local minimizer). *Fix any $\gamma > 0$. Consider the chain precedence graph $g \to r \to a \to b$ with the naive parent penalty*

$$V_{\mathrm{par}}(x) = x_r(1 - x_g) + x_a(1 - x_r) + x_b(1 - x_a).$$

*Define the relaxed objective over $[0,1]^4$ by*

$$\hat{f}(x) := -w^\top x + \gamma V_{\mathrm{par}}(x), \quad x = (x_g, x_r, x_a, x_b),$$

*and choose weights*

$$w_g = w_r = -1, \qquad w_a = w_b = 1.$$

*Then the infeasible binary point*

$$x^\star = (0, 0, 1, 1)$$

*is a strict local minimizer of $\hat{f}$ over $[0, 1]^4$.*

### 3.2. Set Packing and 0–1 Knapsack

The family of packing-type constraints naturally splits into two regimes. When each constraint has right-hand side 1 and 0–1 incidence coefficients, pairwise conflict penalties are the natural square-free construction. By contrast, for genuine knapsack constraints one typically introduces slack variables and starts from a squared residual penalty. We therefore treat these cases separately.

#### 3.2.1. SET PACKING

We first consider the *set packing* class

$$\max \sum_{i=1}^{n} w_i x_i$$
$$\text{s.t.} \sum_{i=1}^{n} A_{k,i} x_i \leq 1, \qquad k = 1, \dots, K, \tag{5}$$
$$x \in \{0, 1\}^n,$$

where $w \in \mathbb{R}_+^n$ and $A \in \{0, 1\}^{K \times n}$. Since each row has right-hand side 1, a violation means that at least two active variables appear in the same constraint. This makes pairwise conflict penalties a natural diagonal-free encoding.

**Examples.**

- **Graph MIS (edge packing).** For a graph $G = (V, E)$, the constraints $x_u + x_v \leq 1$ for each edge $(u, v) \in E$ can be written as $Ax \leq \mathbf{1}$ with one row per edge and two ones per row (Karp, 2009). This is the familiar MIS feasibility condition.
- **$k$-set packing.** Let $U$ be a universe of elements and let $\mathcal{S} = \{S_1, \dots, S_n\}$ be sets with $|S_i| \leq k$. Choosing disjoint sets is equivalent to requiring that for each element $e \in U$, at most one chosen set may contain $e$,

$$\sum_{i:\, e \in S_i} x_i \leq 1.$$

  For $k \geq 3$, this problem is NP-hard (Chan & Lau, 2012).

To construct a QUBO, we use the pairwise conflict penalty

$$V_{\text{sp}}(x) := \sum_{k=1}^{K} \sum_{1 \leq i < j \leq n} A_{k,i} A_{k,j}\, x_i x_j. \tag{6}$$

The corresponding relaxed objective is

$$\hat{f}_{\text{sp}}(x) := -w^\top x + \gamma V_{\text{sp}}(x). \tag{7}$$

The penalty $V_{\text{sp}}(x)$ is diagonal-free with integer coefficients, so Theorem 2.5 implies core binarity whenever $\gamma > \max_i w_i$. Moreover, for binary $x$, the condition $V_{\text{sp}}(x) = 0$ is equivalent to feasibility of (5).

**Proposition 3.4** (Feasibility of local minimizers for set packing). *Assume that any local minimizer of (7) is binary on the core. If $\gamma > \max_i w_i$, then every PGD local minimizer of (7) is feasible for (5).*

The proof follows the local-repairability template: if a binary point violates some row $k$, then at least two active variables appear in that row, and turning off any one of them strictly decreases the penalty (6).

#### 3.2.2. SINGLE-CONSTRAINT 0–1 KNAPSACK AND BINARY-EQUIVALENT CANCELLATION

We now turn to a single knapsack-type constraint

$$\max \sum_{i=1}^{n} w_i x_i$$
$$\text{s.t.} \sum_{i=1}^{n} a_i x_i \leq b, \qquad x \in \{0, 1\}^n, \tag{8}$$

and first focus on the binary-coefficient case $a_i \in \{0, 1\}$.

Unlike set packing, here we encode the inequality by introducing slack bits. Let

$$s(y) := \sum_{\ell=0}^{M} 2^\ell\, y_\ell, \qquad y \in [0, 1]^{M+1},$$

and define the residual

$$r(x, y) := a^\top x + s(y) - b.$$

With sufficiently many bits, any integer slack in $[0, 2^{M+1} - 1]$ is representable. A natural QUBO representation of (8) is

$$\hat{f}(x, y) := -w^\top x + \gamma\, r(x, y)^2. \tag{9}$$

However, the squared residual contains diagonal terms in the core variables, and therefore may admit nonbinary local minimizers in the box relaxation.

To remove these diagonal terms while preserving the objective on binary core points, we consider the following formulation

$$V_{\text{be}}(x, y) := r(x, y)^2 - \sum_{i=1}^{n} a_i x_i^2 + \sum_{i=1}^{n} a_i x_i. \tag{10}$$

We then consider the relaxed optimization problem

$$\min_{(x,y)\in[0,1]^n\times[0,1]^{M+1}} \hat{f}(x,y) := -w^\top x + \gamma V_{\text{be}}(x,y). \quad (11)$$

We say that (10) is *binary-equivalent* to the raw squared penalty if they agree on all binary core assignments.

**Definition 3.5.** A quadratic polynomial $\tilde{V}(z)$ is said to be *binary-equivalent* to $V(z)$ if $\tilde{V}(z) = V(z)$ for all core $x$ encoded in $z = (x,y) \in \{0,1\}^n \times [0,1]^{M+1}$.

In other words, binary-equivalent transformations preserve the value of the objective on the discrete feasible/infeasible core assignments of interest, while allowing us to reshape the relaxed landscape. For (10), the core part is diagonal-free with integer coefficients, so Theorem 2.5 again yields core binarity whenever $\gamma > \max_i w_i$.

**Theorem 3.6** (Feasibility of local minimizers for single-constraint binary-coefficient knapsack). *Assume $a_i \in \{0,1\}$ for all $i$, and assume that any local minimizer of (11) is binary on the core. If $\gamma > \max_i w_i$, then every PGD local minimizer of (11) is feasible for (8).*

The key point is that in the single-constraint binary-coefficient case, an infeasible binary point has positive residual $r \geq 1$, and turning off any active variable in the violated constraint yields a strict penalty decrease. Thus the binary-equivalent cancellation is sufficient in this case.

For general multi-constraint knapsack problems with integer coefficients, binary-equivalent diagonal cancellation is no longer sufficient to guarantee local repairability. We discuss this obstruction and a repair-oriented over-corrected construction in Appendix G. Since the construction does not preserve the binary objective values in the same way as the single-constraint formulation, we treat it as an extension rather than as part of the main theorem pipeline.

**3.3. Traveling Salesman Problem**

This section illustrates how our framework applies to the traveling salesman problem (TSP). We first consider the assignment problem (Lawler, 2001), allowing both multiple cycles and self-loops (standing traveling salesman). We then show that the time-indexed TSP fits an extension of our framework to a quadratic core objective, yielding a relaxed QUBO whose local minimizers are binary and feasible TSP solutions.

Throughout, let $n \geq 3$ and let $c_{ij} \geq 0$ be travel costs.

### 3.3.1. Assignment Problem

We begin with the assignment problem

$$\min_x \sum_{i=1}^n \sum_{j=1}^n c_{ij} x_{ij} \quad (12)$$

$$\text{s.t.} \sum_{j=1}^n x_{ij} = 1 \ \forall i, \quad \sum_{i=1}^n x_{ij} = 1 \ \forall j, \quad x_{ij} \in \{0,1\}.$$

Feasible solutions are permutation matrices. Equivalently, they encode a cycle cover of the cities, where 1-cycles $x_{ii} = 1$ are allowed with $c_{ii} = 0$. We therefore interpret (12) as an assignment model with standing moves.

**Square-free penalty for degree equalities.** Define residuals

$$\rho_i(x) := \sum_{j=1}^n x_{ij} - 1, \qquad \kappa_j(x) := \sum_{i=1}^n x_{ij} - 1.$$

The raw squared penalty $\sum_i \rho_i(x)^2 + \sum_j \kappa_j(x)^2$ contains diagonal core terms $x_{ij}^2$. Since each $x_{ij}$ appears in exactly one row residual and one column residual, its diagonal coefficient in the raw squared penalty is 2. We therefore apply a binary-equivalent diagonal cancellation and define

$$V_{\text{deg}}(x) := \sum_{i=1}^n \rho_i^2(x) + \sum_{j=1}^n \kappa_j^2(x)$$
$$- 2\sum_{i=1}^n \sum_{j=1}^n x_{ij}^2 + 2\sum_{i=1}^n \sum_{j=1}^n x_{ij}. \quad (13)$$

We consider the box-relaxed penalized objective

$$\hat{f}_{\text{deg}}(x) := \sum_{i=1}^n \sum_{j=1}^n c_{ij} x_{ij} + \varepsilon \sum_{i=1}^n \sum_{j=1}^n x_{ij} + \gamma\, V_{\text{deg}}(x).$$
$$(14)$$

where $\varepsilon \sum_{i=1}^n \sum_{j=1}^n x_{ij}$ is added to make every variable appear in the core with a positive coefficient and equals $n\varepsilon$ on every feasible assignments.

**Binarity.** Since $V_{\text{deg}}$ is diagonal-free with integer-valued coefficients on the core variables, Theorem 2.5 implies that every PGD local minimum of (14) is binary on the core provided

$$\gamma > c_{\max}, \qquad c_{\max} := \max_{1 \leq i,j \leq n} c_{ij} + \varepsilon.$$

**Theorem 3.7** (Feasibility of local minimizers for the assignment relaxation). *Assume $c_{ii} = 0 \ \forall i$ and $c_{ij} \geq 0$ for all $i \neq j$ and let $c_{\max} := \max_{i,j} c_{ij} + \varepsilon$. Consider (14) with $V_{\text{deg}}$ defined in (13). Assume $\gamma > c_{\max}$ so that every PGD*

*local minimum has a binary core. Then every PGD local minimum $x^\star$ satisfies the assignment equalities*

$$\sum_{j=1}^n x_{ij}^\star = 1 \ \forall i, \qquad \sum_{i=1}^n x_{ij}^\star = 1 \ \forall j,$$

*and hence is feasible for* (12).

### 3.3.2. TIME-INDEXED TSP AND DIAGONAL-FREE QUADRATIC CORES

We now turn to a formulation that *does* fit our local-correctness framework. Let $x_{i,t} \in \{0,1\}$ indicate that city $i$ is visited at time $t$, with time interpreted cyclically. The time-indexed TSP is

$$\min_x \quad Q(x) := \sum_{t=1}^n \sum_{i=1}^n \sum_{j=1}^n c_{ij}\, x_{i,t}\, x_{j,t+1}$$

$$\text{s.t.} \quad \sum_{i=1}^n x_{i,t} = 1, \qquad t = 1,\dots,n,$$

$$\sum_{t=1}^n x_{i,t} = 1, \qquad i = 1,\dots,n,$$

$$x_{i,t} \in \{0,1\}.$$

Any feasible point is a permutation matrix in the city–time grid, and therefore encodes a Hamiltonian tour.

Define the assignment residuals

$$\rho_t(x) := \sum_{i=1}^n x_{i,t} - 1, \qquad \kappa_i(x) := \sum_{t=1}^n x_{i,t} - 1.$$

As in the assignment model, the raw squared penalty $\sum_t \rho_t(x)^2 + \sum_i \kappa_i(x)^2$ contains diagonal terms $x_{i,t}^2$. Since each variable appears in exactly one time residual and one city residual, we again apply a binary-equivalent diagonal cancellation and define

$$V_{\text{time}}(x) := \sum_{t=1}^n \rho_t(x)^2 + \sum_{i=1}^n \kappa_i(x)^2$$
$$- 2\sum_{i=1}^n \sum_{t=1}^n x_{i,t}^2 + 2\sum_{i=1}^n \sum_{t=1}^n x_{i,t}.$$

As previous, we make every variable appear in the core with a positive coefficient by adding a tiny tie-break term $\varepsilon \sum_{i,t} x_{i,t}$, which is constant over feasible tours. The relaxed objective is

$$\hat{f}_{\text{time}}(x) := Q(x) + \varepsilon \sum_{i=1}^n \sum_{t=1}^n x_{i,t} + \gamma V_{\text{time}}(x), \ \ x \in [0,1]^{n \times n}.$$
$$(15)$$

This formulation no longer has a linear core, but it fits the diagonal-free quadratic-core extension of our theory:

the travel cost $Q(x)$ is bilinear, while the penalty $V_{\text{time}}$ is square-free on the core. The same large-$\gamma$ mechanism then yields binarity and feasibility of local minimizers.

**Theorem 3.8** (Local correctness of the time-indexed TSP relaxation)**.** *Fix $\varepsilon > 0$ and consider* (15)*, then there exists a finite threshold $\gamma_{\text{time}}^\star > 0$, such that for all $\gamma > \gamma_{\text{time}}^\star$, every PGD local minimizer of* (15) *is binary and satisfies the assignment equalities*

$$\sum_{i=1}^n x_{i,t}^\star = 1 \ \forall t, \qquad \sum_{t=1}^n x_{i,t}^\star = 1 \ \forall i.$$

*Remark* 3.9 (Removing the positive-cost assumption). If some $c_{ij} = 0$, replace $c_{ij}$ by $c_{ij} + \varepsilon'$ for any $\varepsilon' > 0$. Since every feasible tour uses exactly $n$ transitions, this adds the constant $n\varepsilon'$ to all feasible tours, but ensures all edge interactions appear in the quadratic core.

## 4. Numerical Experiments

We present two experiments. First, we validate the local-correctness theory on open-pit mining (MineLib (Espinoza et al., 2013)), 0–1 knapsack (kplib (Pisinger, 2005)), and traveling salesman (TSPLIB (Reinelt, 1991)) by comparing naive squared penalties with our theory-guided constructions. Second, we give solver-context results on maximum independent set (MIS), a canonical packing problem covered by our theory. These experiments are intended as empirical context for the relaxed QUBO formulation, not as a comprehensive benchmark against specialized solvers.

### 4.1. Validation of Local-Correctness Theory

Two instances are tested for each problem class, where for each problem we compare a standard (naive) QUBO formulation against our theory-guided construction. Each configuration is run from 10 random initializations, and we report the empirical *binarity rate* and *feasibility rate* of the returned solutions.

**Penalty regimes.** For knapsack and TSP we test two penalty weights: $\gamma_{\text{small}} = \gamma^* + 0.1$ where $\gamma^*$ is the threshold according to Thm 2.8 and $\gamma_{\text{large}} = 10^3\,\gamma_{\text{small}}$, intended to approximate the "$\gamma \to \infty$" regime. We report the resulting ranges of $\gamma_{\text{small}}$ across instances in the corresponding tables.

**Optimizers.** For slack-free QUBOs (open pit and time-indexed TSP), we use PGD, aligned with our theoretical characterization of PGD local minima. For knapsack, slack expansions can make the landscape highly ill-conditioned; we therefore use projected Adam, which more reliably escapes flat first-order traps, and evaluate correctness using the same binarity/feasibility criteria. Since the preconditioning factors are strictly positive, the set of first-order optimality conditions remains unchanged.

*Table 1.* Correctness verification of the relaxed QUBOs of two open pit instances. $\gamma^* \approx 1e16$.

| | Theory-Guided QUBO | | Naive QUBO | |
|---|---|---|---|---|
| $\gamma^* + 0.1$ | Bin: | 100% (20/20) | Bin: | 60% (12/20) |
| | Feas: | 100% (20/20) | Feas: | 0% (0/20) |

*Table 2.* Correctness verification of the relaxed QUBOs of two knapsack instances. $\gamma^* \approx 1000$.

| | Theory-Guided QUBO | | Naive QUBO | |
|---|---|---|---|---|
| $\gamma_{\text{small}}$ | Bin: | 100% (20/20) | Bin: | 0% (0/20) |
| | Feas: | 100% (20/20) | Feas: | 100% (20/20) |
| $\gamma_{\text{large}}$ | Bin: | 100% (20/20) | Bin: | 0% (0/20) |
| | Feas: | 100% (20/20) | Feas: | 100% (20/20) |

**Open-pit mining.** We contrast the theory-guided ancestor form QUBO with the commonly used parent form QUBO. These MineLib instances contain "air" blocks with extremely negative values that are typically removed during preprocessing. We intentionally avoid such preprocessing and run on the raw block models to stress-test robustness. Even at this very large theoretical threshold, the ancestor formulation succeeds on the raw block model while the parent formulation fails. Because the raw penalty scale is already dominated by these large-magnitude dummy blocks, we focus on formulation comparison rather than a separate "very large $\gamma$" ablation.

**Knapsack and TSP.** Across all tested instances, our proposed QUBOs achieve binary and feasible solutions under both $\gamma_{\text{small}}$ and $\gamma_{\text{large}}$. In contrast, the naive QUBOs can converge to fractional and/or infeasible local minima, and increasing $\gamma$ alone does not reliably remove these failures.

### 4.2. Maximum Independent Set: Practical Context on a Canonical NP-hard Packing Problem

We next study the maximum independent set (MIS) problem as a canonical NP-hard packing problem. This experiment complements the theory developed earlier for packing-type relaxations. The goal here is not to present a new specialized MIS heuristic, but to show that the relaxed-QUBO formulation studied in this paper supports practically competitive local optimization when solved by PGD.

Table 4 reports the average independent-set size over 8 Erdős–Rényi instances per setting under a 5-minute budget. PGD returns larger independent sets than Gurobi across all tested sizes and densities. Table 5 compares PGD with lightweight heuristics on a representative subset. The results show that PGD is competitive with stochastic local search (SLS) and GRASP, and that batched GPU restarts improve the short-budget regime. Here SLS denotes stochastic perturbation (ratio 25%) followed by greedy repair, while GRASP denotes a semi-greedy randomized construction followed by local search.

*Table 3.* Correctness verification for relaxed time-indexed TSP QUBOs under two penalty regimes for two TSP instances. $\gamma^* \approx \{1e5, 3e6\}$ each for one of the two instances.

| | Theory-Guided QUBO | | Naive QUBO | |
|---|---|---|---|---|
| $\gamma_{\text{small}}$ | Bin: | 100% (20/20) | Bin: | 0% (0/20) |
| | Feas: | 100% (20/20) | Feas: | 0% (0/20) |
| $\gamma_{\text{large}}$ | Bin: | 100% (20/20) | Bin: | 5% (1/20) |
| | Feas: | 100% (20/20) | Feas: | 0% (0/20) |

*Table 4.* Average MIS size under a fixed 5-minute budget on 8 Erdős–Rényi instances for each $(n, p)$ setting. PGD and Gurobi are run under the same wall-clock budget.

| $(n,p)$ | PGD (0.3) | Gurobi (0.3) | PGD (0.5) | Gurobi (0.5) | PGD (0.7) | Gurobi (0.7) |
|---|---|---|---|---|---|---|
| 2000 | 27 | 21.875 | 14.25 | 13.125 | 10 | 8.375 |
| 4000 | 28.5 | 20.5 | 16 | 11.625 | 10.5 | 7.375 |
| 6000 | 30.25 | 21.75 | 17.375 | 12.25 | 10.5 | 8 |
| 8000 | 30.875 | 23.25 | 17.375 | 13.25 | 10.75 | 7.75 |
| 10000 | 30.625 | 22.875 | 17 | 13.5 | 10.625 | 8.25 |
| 20000 | 32.5 | 24.5 | 17.875 | 14 | 11 | 8.625 |
| 30000 | 33.625 | 26.25 | 18.75 | 14.25 | 11.25 | 9.125 |
| 40000 | 34.125 | 26.75 | 18.875 | 15.125 | 11.625 | 9.625 |

*Table 5.* Representative MIS comparisons under 5-minute and 30-second budgets. Rows are ordered from smaller- to larger-MIS regimes, rather than by graph size alone. "PGD (1GPU Parallel)" denotes batched parallel restarts optimized simultaneously on one GPU.

| 5 minutes | | | | |
|---|---|---|---|---|
| $(n,p)$ | PGD | Gurobi | SLS | GRASP |
| $(8000, 0.7)$ | 10.75 | 7.75 | 10.5 | 11.125 |
| $(4000, 0.5)$ | 16 | 11.625 | 16.125 | 17 |
| $(2000, 0.3)$ | 27 | 21.875 | 25.625 | 27.125 |
| $(4000, 0.3)$ | 28.5 | 20.5 | 28.25 | 29 |
| $(8000, 0.3)$ | 30.875 | 23.25 | 30.375 | 30.625 |
| $(3000, 1/30)$ | 198.5 | 142.75 | 194.75 | 186.125 |

| 30 seconds | | | | |
|---|---|---|---|---|
| $(n,p)$ | PGD (1GPU Parallel) | PGD | SLS | GRASP |
| $(8000, 0.7)$ | 11.125 | 10.125 | 10.375 | 11 |
| $(4000, 0.5)$ | 16.375 | 15.75 | 15.75 | 16.25 |
| $(2000, 0.3)$ | 26.75 | 25.75 | 25.625 | 26.75 |
| $(4000, 0.3)$ | 29 | 28.125 | 28.125 | 28.125 |
| $(8000, 0.3)$ | 31 | 29.625 | 29.875 | 30 |
| $(3000, 1/30)$ | 200.75 | 191.5 | 191.75 | 183.5 |

## 5. Conclusion

We studied when solving box-relaxed QUBOs with first-order methods can be *certifiably correct* for the underlying binary constrained problem. Under diagonal-free penalty structure, we showed that sufficiently large penalties rule out fractional local minima on the core variables, yielding binarity guarantees for box-local minimizers. We then introduced a local-repairability condition for constraint penalties, providing explicit thresholds under which any box-local minimizer is not only binary but also feasible. A practical design principle is highlighted: formulation choice is part of the theory, and selecting a certifiable formulation can be more important than tuning the penalty.

## Acknowledgment

This work was supported by the Defense Advanced Research Projects Agency (DARPA) under Cooperative Agreement No. HR0011-25-2-0021.

## Impact Statement

This paper provides theoretical conditions and explicit penalty thresholds under which box-constrained local minimizers of relaxed QUBO objectives are guaranteed to be binary and feasible for the underlying discrete constraints. The main positive impact is improved reliability and interpretability of gradient-based heuristics for QUBO models, including clarifying how formulation choices affect correctness.

The results are purely theoretical and do not guarantee global optimality. Potential negative impact comes from overinterpreting the guarantees or using excessively large penalties that can harm optimization or numerical stability; practitioners should treat the theory as a correctness check under stated assumptions and incorporate domain-specific constraints and oversight in any real deployment.

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

# Appendix

In the appendix, we collect full proofs and technical details that are omitted from the main text for readability. The appendix is organized as follows:

1. **Appendix A** provides the proof of Lemma 2.4, establishing the interior-structure property used throughout the binarity analysis.

2. **Appendix B** provides the proof of Theorem 2.5, which gives the core binarity guarantee for diagonal-free integer-coefficient penalties.

3. **Appendix C** provides the proof of Theorem 2.7, proving feasibility of local minimizers under the local-repairability condition and a sufficiently large penalty.

4. **Appendix D** provides the proof of Proposition 3.1, establishing feasibility (via repair directions) for the *ancestor* formulation in open-pit mining.

5. **Appendix E** provides the proof of Lemma 3.3, giving a concrete failure mode of the naive *parent* formulation by constructing an infeasible strict local minimizer.

6. **Appendix F** provides the proof of Theorem 3.6 for the single-constraint knapsack setting, deriving an explicit penalty threshold that enforces feasibility.

7. **Appendix G** discusses the extension from single-constraint knapsack to multi-constraint knapsack.

8. **Appendix H** provides the proof of Theorem 3.7 for the assignment problem.

9. **Appendix I** presents the time-indexed TSP extension (quadratic/bilinear travel-cost core): the square-free penalty construction, the interior-structure lemma, and the resulting core-binarity and feasibility guarantees.

## A. Proof of Lemma 2.4

*Proof.* Fix a local minimizer $z^\star$ of $\hat{f}(z) = w^\top z + \gamma V(z)$ over the box $[0,1]^N$ and let $J := J(z^\star) = \{i : 0 < z_i^\star < 1\}$. Assume toward a contradiction that there exist distinct indices $i, j \in J$ with $Q_{ij} \neq 0$ and at least one of $i$ and $j$ is a core variable. Since $(z_i^\star, z_j^\star) \in (0,1)^2$, there exists $\varepsilon > 0$ such that for all $(s,t) \in \mathbb{R}^2$ with $\|(s,t)\|_2 \leq \varepsilon$,

$$z^\star + se_i + te_j \in [0,1]^N.$$

Define the two–variable restriction

$$\phi(s,t) := \hat{f}(z^\star + se_i + te_j).$$

Then $(0,0)$ is a (unconstrained) local minimizer of the $C^2$ function $\phi$, so the $2 \times 2$ Hessian $H := \nabla^2\phi(0,0)$ must be positive semidefinite.

Because $w^\top z$ is linear, it contributes no second derivatives. For the penalty, the only term in $V$ that creates a mixed second derivative between coordinates $i$ and $j$ is $Q_{ij}z_i z_j$. Hence, the $(i,j)$ block of $\nabla^2 V(z^\star)$ has the form

$$\nabla^2_{(i,j)} V(z^\star) = \begin{bmatrix} 2Q_{ii} & Q_{ij} \\ Q_{ij} & 2Q_{jj} \end{bmatrix},$$

where we interpret $Q_{kk} = 0$ for those indices on which $V$ has no diagonal term (in particular, for core indices under the diagonal-free assumption). Therefore

$$H = \gamma \begin{bmatrix} 2Q_{ii} & Q_{ij} \\ Q_{ij} & 2Q_{jj} \end{bmatrix}.$$

Its determinant is

$$\det(H) = \gamma^2 \big(4Q_{ii}Q_{jj} - Q_{ij}^2\big) \leq -\gamma^2 Q_{ij}^2 < 0,$$

so $H$ is indefinite, contradicting the second-order necessary condition at $(0,0)$. We conclude that no interior core coordinate can interact with another interior coordinate. $\square$

## B. Proof of Thm 2.5

*Proof.* Let $z^\star$ be a PGD local minimizer of $\hat{f}(z) = w^\top z + \gamma V(z)$ over $[0,1]^N$ and recall the core set $\mathcal{C} = \{i : w_i \neq 0\}$. Fix any core index $i \in \mathcal{C}$. Suppose toward a contradiction that $0 < z_i^\star < 1$.

By Lemma 2.4, the interior set $J(z^\star)$ is a core independent set of the interaction graph induced by the off–diagonal coefficients $Q_{ij}$. In particular, every index $j$ with $Q_{ij} \neq 0$ must satisfy $z_j^\star \in \{0,1\}$.

Compute the partial derivative of $V$ at $z^\star$ in the $i$th coordinate:

$$\partial_i V(z^\star) = \sum_{j \neq i} Q_{ij} z_j^\star + d_i.$$

Because $Q_{ij}, d_i \in \mathbb{Z}$ and each $z_j^\star \in \{0,1\}$ whenever $Q_{ij} \neq 0$, we have $\partial_i V(z^\star) \in \mathbb{Z}$.

Since $z_i^\star$ is interior, the first-order stationarity condition for box constraints reduces to $\partial_i \hat{f}(z^\star) = 0$, i.e.,

$$0 = \partial_i \hat{f}(z^\star) = w_i + \gamma \, \partial_i V(z^\star).$$

Let $k := \partial_i V(z^\star) \in \mathbb{Z}$. Then $w_i = -\gamma k$ and thus $|w_i| = |\gamma|\,|k|$. Under the assumption $|\gamma| > \max_{i \in \mathcal{C}} |w_i|$, we have $|\gamma| > |w_i|$; hence the equality $|w_i| = |\gamma|\,|k|$ forces $k = 0$. But then $0 = w_i + \gamma k = w_i$, contradicting $i \in \mathcal{C}$ (i.e., $w_i \neq 0$). Therefore no core coordinate can be interior, and $z_i^\star \in \{0,1\}$ for all $i \in \mathcal{C}$. $\square$

## C. Proof of Thm 2.7

*Proof.* Let $z^\star$ be a local minimizer of

$$\hat{f}(z) = w^\top z + \gamma V(z)$$

over $[0,1]^N$. Assume for contradiction that the core of $z^\star$ is infeasible.

By assumption (i), every local minimizer has binary core, so the core of $z^\star$ is binary and infeasible. Hence assumption (ii) applies: there exists a repair direction

$$d \in \mathcal{D} \cap T(z^\star)$$

such that

$$\langle \nabla V(z^\star), d \rangle \leq -\delta.$$

By the definition of $L$,

$$\langle w, d \rangle \leq L.$$

Therefore

$$\langle \nabla \hat{f}(z^\star), d \rangle = \langle w, d \rangle + \gamma \langle \nabla V(z^\star), d \rangle \leq L - \gamma \delta.$$

If $\gamma > L/\delta$, then

$$\langle \nabla \hat{f}(z^\star), d \rangle < 0.$$

Since $d \in T(z^\star)$, the standard first-order necessary condition for a local minimizer over a box implies

$$\langle \nabla \hat{f}(z^\star), d \rangle \geq 0,$$

a contradiction. Hence the core of $z^\star$ must be feasible. $\square$

## D. Proof of Proposition 3.1

*Proof.* Assume core binarity holds. Let $x \in \{0,1\}^n$ be an infeasible binary point. Let $S = \{i : x_i = 1\}$ be the set of extracted blocks. Since $x$ is infeasible, there exists a block $c \in S$ that violates a precedence constraint; that is, there is an ancestor $\alpha \in Anc(c)$ such that $x_\alpha = 0$. This implies $\sum_{\alpha \in Anc(c)} (1 - x_\alpha) \geq 1$.

We select a specific violating node $c^*$ to repair. Let $V_{viol} = \{c \in S : \exists \alpha \in Anc(c), x_\alpha = 0\}$ be the set of extracted blocks with missing ancestors. Choose $c^*$ to be a *maximal* element of $V_{viol}$ with respect to the graph topology (i.e., $c^*$ has no

descendants in $V_{viol}$). In fact, we can choose $c^*$ to be maximal in $S$; if a node is in $S$, any descendant in $S$ implies the ancestor is satisfied relative to the descendant, but we simply need a node $c^* \in S$ such that no descendant of $c^*$ is in $S$. If $x$ is infeasible, we can always find a "deepest" active node $c^*$ that violates ancestry constraints (or whose ancestors do) such that turning $c^*$ off does not create new violations for its descendants (because it has none active).

Consider the direction $d = -e_{c^*} \in D \cap T(x)$, which corresponds to un-extracting block $c^*$. The gradient of the penalty is:

$$\partial_i V_{\mathrm{anc}}(x) = \sum_{\alpha \in Anc(i)} (1 - x_\alpha) - \sum_{k : i \in Anc(k)} x_k.$$

Evaluating this at $c^*$: 1. The ancestor term $\sum_{\alpha \in Anc(c^*)}(1 - x_\alpha) \geq 1$ because $c^*$ is infeasible. 2. The descendant term $\sum_{k : c^* \in Anc(k)} x_k = 0$ because $c^*$ was chosen such that no active descendants exist (or we recursively remove them). Specifically, in the ancestor formulation, removing a node $c$ removes the requirement for its ancestors to be present for $c$, decreasing the penalty. The only penalty increase could come from $c$ being an ancestor to some other active $k$. However, if we choose $c^*$ such that it has no active descendants, this term is 0.

Thus, $\partial_{c^*} V_{\mathrm{anc}}(x) \geq 1$. The directional derivative is:

$$\langle \nabla V_{\mathrm{anc}}(x), d \rangle = -\partial_{c^*} V_{\mathrm{anc}}(x) \leq -1.$$

This confirms local repairability with margin $\delta = 1$. The linear gain bound is $L = \sup_{d \in D} \langle -w, d \rangle = \sup_i w_i$. By Theorem 2.7, any $\gamma > (\max_i w_i)/1$ guarantees feasibility. $\qquad \square$

## E. Proof of Lemma 3.3

*Proof.* First note $x^\star$ is infeasible since $r \to a$ is violated: $x_a^\star(1 - x_r^\star) = 1$. Also

$$V_{\mathrm{par}}(x^\star) = x_a^\star(1 - x_r^\star) + x_b^\star(1 - x_a^\star) = 1.$$

*Step 1: The parent penalty is first-order flat at $x^\star$.* A direct differentiation gives

$$\partial_g V_{\mathrm{par}} = -x_r, \quad \partial_r V_{\mathrm{par}} = (1 - x_g) - x_a, \quad \partial_a V_{\mathrm{par}} = (1 - x_r) - x_b, \quad \partial_b V_{\mathrm{par}} = (1 - x_a),$$

hence $\nabla V_{\mathrm{par}}(x^\star) = 0$.

*Step 2: Tangent-cone directions have a uniform first-order increase from the linear term.* The tangent cone of the box at $x^\star$ is

$$T(x^\star) = \{h \in \mathbb{R}^4 : h_g \geq 0, \ h_r \geq 0, \ h_a \leq 0, \ h_b \leq 0\}.$$

For any $h \in T(x^\star)$,

$$\langle \nabla(-w^\top x)\big|_{x = x^\star}, h \rangle = \langle -w, h \rangle = (-w_g)h_g + (-w_r)h_r + (-w_a)h_a + (-w_b)h_b = h_g + h_r - h_a - h_b = \|h\|_1.$$

In particular, $\langle -w, h \rangle \geq \|h\|_2$ for all $h \in T(x^\star)$.

*Step 3: The penalty changes only at second order near $x^\star$.* Since $V_{\mathrm{par}}$ is a quadratic polynomial, Taylor's theorem is exact:

$$V_{\mathrm{par}}(x^\star + h) - V_{\mathrm{par}}(x^\star) = \langle \nabla V_{\mathrm{par}}(x^\star), h \rangle + \tfrac{1}{2}h^\top H h = \tfrac{1}{2}h^\top H h,$$

where the Hessian $H$ is constant with off-diagonal entries only on the edges $(g, r), (r, a), (a, b)$ and zeros on the diagonal. In particular, $\|H\|_2 \leq 2$ (each row has at most two nonzeros of magnitude 1), so

$$V_{\mathrm{par}}(x^\star + h) - V_{\mathrm{par}}(x^\star) \geq -\tfrac{1}{2}\|H\|_2\|h\|_2^2 \geq -\|h\|_2^2.$$

*Step 4: Strict local minimality.* Combining the above,

$$\hat{f}(x^\star + h) - \hat{f}(x^\star) = \langle -w, h \rangle + \gamma\big(V_{\mathrm{par}}(x^\star + h) - V_{\mathrm{par}}(x^\star)\big) \geq \|h\|_2 - \gamma\|h\|_2^2.$$

Thus for any nonzero feasible perturbation with $\|h\|_2 < 1/\gamma$ we have $\hat{f}(x^\star + h) > \hat{f}(x^\star)$, proving that $x^\star$ is a *strict* local minimizer over $[0, 1]^4$.

Finally, note this also aligns with the box first-order necessary condition: at a local minimizer, every feasible direction $d \in T(x^\star)$ must satisfy $\langle \nabla \hat{f}(x^\star), d \rangle \geq 0$; here $\nabla \hat{f}(x^\star) = -w$ and the inequality holds strictly for all $d \in T(x^\star) \setminus \{0\}$. $\qquad \square$

## F. Proof of Thm 3.6

*Proof.* Let $(x^\star, y^\star)$ be a PGD local minimizer of (11). By assumption, its core $x^\star$ is binary. Assume for contradiction that $(x^\star, y^\star)$ is infeasible for (8). Then

$$a^\top x^\star > b.$$

Since $a^\top x^\star$ and $b$ are integers, we have

$$a^\top x^\star \geq b + 1.$$

Because the slack encoding satisfies $s(y^\star) \geq 0$, the residual obeys

$$r(x^\star, y^\star) = a^\top x^\star + s(y^\star) - b \geq 1.$$

Since $x^\star$ is binary and infeasible, there exists an index $i$ such that

$$x_i^\star = 1 \qquad \text{and} \qquad a_i = 1.$$

Consider the feasible direction

$$d := (-e_i, 0),$$

which turns off this active core variable and leaves the slack variables unchanged. Because $x_i^\star = 1$, we have $d \in T(x^\star, y^\star)$.

Now

$$V_{\text{be}}(x, y) = r(x, y)^2 - \sum_{j=1}^n a_j x_j^2 + \sum_{j=1}^n a_j x_j,$$

so for any $i$,

$$\frac{\partial V_{\text{be}}}{\partial x_i}(x, y) = 2a_i\, r(x, y) - 2a_i x_i + a_i.$$

Evaluating at $(x^\star, y^\star)$ and using $a_i = 1$, $x_i^\star = 1$, and $r(x^\star, y^\star) \geq 1$, we obtain

$$\frac{\partial V_{\text{be}}}{\partial x_i}(x^\star, y^\star) = 2\, r(x^\star, y^\star) - 1 \geq 1.$$

Hence

$$\langle \nabla V_{\text{be}}(x^\star, y^\star), d \rangle = -\frac{\partial V_{\text{be}}}{\partial x_i}(x^\star, y^\star) \leq -1.$$

So the penalty is locally repairable with margin $\delta = 1$ relative to the direction set

$$\mathcal{D} := \{(-e_i, 0) : i = 1, \ldots, n\}.$$

For the linear term in (11),

$$\langle (-w, 0), d \rangle = w_i \leq \max_j w_j.$$

Thus the constant in Theorem 2.7 satisfies

$$L \leq \max_j w_j.$$

If $\gamma > \max_j w_j$, then in particular $\gamma > L/\delta = L$, since $\delta = 1$. Therefore Theorem 2.7 implies that every PGD local minimizer of (11) is feasible for (8). $\qquad\square$

## G. Multi-constraint Knapsack

We consider the general multi-constraint knapsack problem

$$\max \sum_{i=1}^n w_i x_i \tag{16}$$
$$\text{s.t. } Ax \leq b, \qquad A \in \mathbb{Z}_{>0}^{K \times n}, \quad b \in \mathbb{Z}_{>0}^K, \quad x \in \{0, 1\}^n.$$

As before, we introduce an independent block of slack bits for each constraint:

$$s_k(y^{(k)}) := \sum_{\ell=0}^{M_k} 2^\ell y_\ell^{(k)}, \quad r_k(x, y^{(k)}) := A_{k,:}x + s_k(y^{(k)}) - b_k.$$

A natural squared-residual construction is then

$$\sum_{k=1}^{K} r_k(x, y^{(k)})^2.$$

For general coefficients and multiple constraints, however, binary-equivalent diagonal cancellation is no longer enough to guarantee that every infeasible binary point admits a descent direction. Indeed, the cancellation

$$V_{\text{be}}^k(x, y) := r_k(x, y^{(k)})^2 - \sum_{i=1}^{n} A_{k,i}^2 (x_i^2 - x_i) \tag{17}$$

removes the core diagonal terms and preserves the value on binary points, but creates local minimizers at infeasible points.

This motivates the over-corrected penalty

$$V_{\text{over}}^k(x, y) := r_k(x, y^{(k)})^2 - \sum_{i=1}^{n} A_{k,i}^2 (x_i^2 - 2x_i), \tag{18}$$

and the aggregated objective

$$\hat{f}(x, y) := -w^\top x + \gamma \sum_{k=1}^{K} V_{\text{over}}^k(x, y). \tag{19}$$

Relative to the binary-equivalent correction (17), the over-correction (18) reshapes the relaxed landscape to favor one-bit repairs at violated constraints, and feasibility follows. In particular, at an infeasible binary point, the derivative of the $k$th bracket with respect to an active variable $x_i = 1$ contains the term $2A_{k,i}r_k(x, y^{(k)})$, which is strictly positive whenever the $k$th constraint is violated. Thus turning off a variable in a violated row becomes a natural repair move, whereas the binary-equivalent correction may admit flat or adverse first-order behavior.

We emphasize the distinction between the two constructions: binary-equivalent cancellation preserves the discrete objective values on binary points, while over-correction is introduced specifically to improve local repairability in the relaxation.

## H. Proof of Thm 3.7

*Proof.* Work with the objective in (14), and let $\tilde{c}_{ij} := c_{ij} + \varepsilon$. Let $x^\star$ be a PGD local minimum of $\hat{f}_{\text{deg}}$ over the box. By the assumed bound $\gamma > c_{\max}$ and Theorem 2.5, the core is binary, so

$$x_{ij}^\star \in \{0, 1\} \qquad \forall\, i, j.$$

Define the row and column residuals

$$\rho_i(x) := \sum_{j=1}^{n} x_{ij} - 1, \qquad \kappa_j(x) := \sum_{i=1}^{n} x_{ij} - 1.$$

Since $x^\star$ is binary, every $\rho_i(x^\star)$ and $\kappa_j(x^\star)$ is an integer.

A direct differentiation of (13) gives, for every pair $(i, j)$,

$$\frac{\partial V_{\text{deg}}}{\partial x_{ij}}(x) = 2\rho_i(x) + 2\kappa_j(x) - 4x_{ij} + 2. \tag{20}$$

Because $x^\star$ is a local minimum over the box, the coordinatewise first-order conditions are

$$x_{ij}^\star = 1 \Rightarrow \frac{\partial \hat{f}_{\text{deg}}}{\partial x_{ij}}(x^\star) \leq 0, \qquad x_{ij}^\star = 0 \Rightarrow \frac{\partial \hat{f}_{\text{deg}}}{\partial x_{ij}}(x^\star) \geq 0. \tag{21}$$

Indeed, when $x_{ij}^\star = 1$, the direction $-e_{ij}$ is feasible; when $x_{ij}^\star = 0$, the direction $+e_{ij}$ is feasible.

**Step 1: no row can have out-degree at least** $2$. Assume for contradiction that there exists a row $i$ with

$$\sum_{j=1}^{n} x_{ij}^{\star} \geq 2,$$

equivalently $\rho_i(x^{\star}) \geq 1$. Choose any $j$ such that $x_{ij}^{\star} = 1$. Then the column sum of column $j$ is at least 1, so

$$\kappa_j(x^{\star}) \geq 0.$$

Using (21) and (20) at this selected entry,

$$0 \geq \frac{\partial \hat{f}_{\mathrm{deg}}}{\partial x_{ij}}(x^{\star}) = \tilde{c}_{ij} + \gamma\Big(2\rho_i(x^{\star}) + 2\kappa_j(x^{\star}) - 4 \cdot 1 + 2\Big),$$

that is,

$$0 \geq \tilde{c}_{ij} + 2\gamma\big(\rho_i(x^{\star}) + \kappa_j(x^{\star}) - 1\big).$$

Since $\tilde{c}_{ij} > 0$, this implies

$$\rho_i(x^{\star}) + \kappa_j(x^{\star}) - 1 < 0.$$

Because the left-hand side is an integer, in fact

$$\rho_i(x^{\star}) + \kappa_j(x^{\star}) - 1 \leq -1, \qquad \text{so} \qquad \rho_i(x^{\star}) + \kappa_j(x^{\star}) \leq 0.$$

But $\rho_i(x^{\star}) \geq 1$ and $\kappa_j(x^{\star}) \geq 0$, hence

$$\rho_i(x^{\star}) + \kappa_j(x^{\star}) \geq 1,$$

a contradiction. Therefore every row has out-degree at most 1.

**Step 2: no row can have out-degree** $0$. Assume for contradiction that there exists a row $i$ with

$$\sum_{j=1}^{n} x_{ij}^{\star} = 0,$$

equivalently $\rho_i(x^{\star}) = -1$. By Step 1, every row has out-degree at most 1. Hence the total number of selected edges is at most $n - 1$:

$$\sum_{i=1}^{n}\sum_{j=1}^{n} x_{ij}^{\star} \leq n - 1.$$

Therefore the total column sum is also at most $n - 1$, so at least one column $\ell$ must have in-degree 0:

$$\sum_{i=1}^{n} x_{i\ell}^{\star} = 0, \qquad \text{equivalently} \qquad \kappa_\ell(x^{\star}) = -1.$$

For this pair $(i, \ell)$ we have $x_{i\ell}^{\star} = 0$. Applying (21) and (20),

$$0 \leq \frac{\partial \hat{f}_{\mathrm{deg}}}{\partial x_{i\ell}}(x^{\star}) = \tilde{c}_{i\ell} + \gamma\Big(2(-1) + 2(-1) - 4 \cdot 0 + 2\Big) = \tilde{c}_{i\ell} - 2\gamma.$$

But $\tilde{c}_{i\ell} \leq c_{\max}$ and $\gamma > c_{\max}$ imply

$$\tilde{c}_{i\ell} - 2\gamma < c_{\max} - 2\gamma < 0,$$

a contradiction. Hence no row can have out-degree 0.

**Conclusion for rows.** Every row has out-degree neither $\geq 2$ nor 0, hence every row has out-degree exactly 1:

$$\sum_{j=1}^{n} x_{ij}^{\star} = 1 \qquad \forall i.$$

**Conclusion for columns.** The argument is symmetric in rows and columns. Equivalently, one may repeat the above proof with the roles of $\rho$ and $\kappa$ interchanged. Thus every column has in-degree exactly 1:

$$\sum_{i=1}^{n} x_{ij}^{\star} = 1 \qquad \forall j.$$

Therefore $x^{\star}$ satisfies all assignment equalities and is feasible for (12). $\qquad\square$

# I. Time-indexed TSP (quadratic objective)

This appendix section records a quadratic-core extension of the main framework: the time-indexed formulation of TSP has a bilinear travel-cost objective. We show that, after a square-free diagonal cancellation of the assignment penalties and a small tie-break, the same binarity/feasibility mechanism applies.

## I.1. Problem and square-free penalty

Let $x_{i,t} \in \{0,1\}$ indicate that city $i \in \{1, \ldots, n\}$ is visited at time $t \in \{1, \ldots, n\}$. We interpret time cyclically: $t + 1$ is modulo $n$ (equivalently, define $x_{\cdot,n+1} := x_{\cdot,1}$ and $x_{\cdot,0} := x_{\cdot,n}$). The integer program is

$$
\begin{aligned}
\min_{x} \quad & Q(x) := \sum_{t=1}^{n} \sum_{i=1}^{n} \sum_{j=1}^{n} c_{ij}\, x_{i,t}\, x_{j,t+1} \\
\text{s.t.} \quad & \sum_{i=1}^{n} x_{i,t} = 1, \qquad t = 1, \ldots, n, \\
& \sum_{t=1}^{n} x_{i,t} = 1, \qquad i = 1, \ldots, n, \\
& x_{i,t} \in \{0,1\}, \qquad i = 1, \ldots, n,\ t = 1, \ldots, n.
\end{aligned}
\tag{22}
$$

Any feasible $x$ is a permutation matrix: it selects exactly one city per time and exactly one time per city, and thus encodes a Hamiltonian tour.

**Square-free penalty for assignment equalities.** Define residuals

$$\rho_t(x) := \sum_{i=1}^{n} x_{i,t} - 1, \qquad \kappa_i(x) := \sum_{t=1}^{n} x_{i,t} - 1.$$

The raw squared penalty $\sum_t \rho_t(x)^2 + \sum_i \kappa_i(x)^2$ contains diagonal terms $x_{i,t}^2$. Since each $x_{i,t}$ appears in exactly one $\rho_t$ and one $\kappa_i$, its diagonal coefficient in the raw squared penalty is 2, so we apply a binary-equivalent diagonal cancellation and define

$$V_{\text{time}}(x) := \sum_{t=1}^{n} \rho_t(x)^2 + \sum_{i=1}^{n} \kappa_i(x)^2 \; - \; 2 \sum_{i=1}^{n} \sum_{t=1}^{n} x_{i,t}^2 \; + \; 2 \sum_{i=1}^{n} \sum_{t=1}^{n} x_{i,t}. \tag{23}$$

By construction, $V_{\text{time}}$ is quadratic and has *zero diagonal* in the variables $x_{i,t}$.

**Relaxed penalized objective with a tie-break.** Fix a small $\varepsilon > 0$ and consider the box relaxation

$$\min_{x \in [0,1]^{n^2}} \hat{f}_{\text{time}}(x) := Q(x) \; + \; \varepsilon \sum_{i=1}^{n} \sum_{t=1}^{n} x_{i,t} \; + \; \gamma\, V_{\text{time}}(x). \tag{24}$$

On the feasible set of (22) we have $\sum_{i,t} x_{i,t} = n$, so the tie-break term $\varepsilon \sum_{i,t} x_{i,t}$ is constant and does not re-rank feasible tours.

### I.2. Useful identities and bounds

Assume $c_{ij} \geq 0$. Differentiating $Q$ under cyclic time gives, for each $(i, t)$,

$$\frac{\partial Q}{\partial x_{i,t}}(x) = \sum_{j=1}^{n} c_{ij}\, x_{j,t+1} \; + \; \sum_{k=1}^{n} c_{ki}\, x_{k,t-1} \; \geq 0. \tag{25}$$

Hence, for $x \in [0,1]^{n^2}$,

$$0 \leq \frac{\partial Q}{\partial x_{i,t}}(x) \leq \sum_{j=1}^{n} c_{ij} + \sum_{k=1}^{n} c_{ki} \quad \Rightarrow \quad \|\nabla Q(x)\|_{\infty} \leq C_{\max}, \tag{26}$$

where

$$C_{\max} := \max_{i \in \{1,\dots,n\}} \Big( \sum_{j=1}^{n} c_{ij} + \sum_{k=1}^{n} c_{ki} \Big). \tag{27}$$

Differentiating (23) yields, for every $(i, t)$,

$$\frac{\partial V_{\text{time}}}{\partial x_{i,t}}(x) = 2\rho_t(x) + 2\kappa_i(x) - 4x_{i,t} + 2. \tag{28}$$

### I.3. A penalty-structure lemma

Let $J(x) := \{(i, t) : 0 < x_{i,t} < 1\}$ be the interior index set.

**Lemma I.1** (Row/column interior independence for $V_{\text{time}}$). *Let $x^{\star}$ be a local minimizer of $\hat{f}_{\text{time}}$ over $[0,1]^{n^2}$. Then $J(x^{\star})$ contains no two indices in the same time slice or the same city:*

$$(i,t), (i',t) \in J(x^{\star}) \Rightarrow i = i', \qquad (i,t), (i,t') \in J(x^{\star}) \Rightarrow t = t'.$$

*Equivalently: for each fixed $t$, at most one $x^{\star}_{i,t}$ is interior; and for each fixed $i$, at most one $x^{\star}_{i,t}$ is interior.*

*Proof.* We prove the time-slice claim; the city claim can be proved in a similar fashion. Assume for contradiction that $(i, t)$ and $(i', t)$ are both interior with $i \neq i'$. Consider the restriction of $\hat{f}_{\text{time}}$ to the 2D affine subspace $x^{\star} + s e_{i,t} + r e_{i',t}$ for small $(s, r)$. Because $x^{\star}_{i,t}, x^{\star}_{i',t} \in (0, 1)$, for sufficiently small $(s, r)$ this stays in the box.

The penalty $V_{\text{time}}$ contains the term $\rho_t(x)^2$ where $\rho_t(x) = \sum_k x_{k,t} - 1$. Hence $\partial^2 \rho_t(x)^2 / \partial x_{i,t} \partial x_{i',t} = 2$ for $i \neq i'$. The diagonal cancellation in (23) removes only diagonal terms in $x_{k,t}^2$ and does not change this mixed second derivative. Therefore the $2 \times 2$ Hessian block of $V_{\text{time}}$ on coordinates $(x_{i,t}, x_{i',t})$ has the form

$$\nabla^2_{(i,i'),t} V_{\text{time}} = \begin{bmatrix} 0 & 2 \\ 2 & 0 \end{bmatrix},$$

which is indefinite.

The linear tie-break term $\varepsilon \sum_{i,t} x_{i,t}$ contributes no second derivatives. Moreover, the bilinear travel cost $Q(x)$ has no squared terms, so it contributes zero diagonal entries to this two-dimensional Hessian block. For two variables in the same time slice, $x_{i,t}$ and $x_{i',t}$, the travel cost $Q$ does not couple them directly, since $Q$ only couples adjacent time slices. Hence the Hessian block of the full objective on these two coordinates is

$$\gamma \begin{bmatrix} 0 & 2 \\ 2 & 0 \end{bmatrix},$$

which has determinant $-4\gamma^2 < 0$. This contradicts the second-order necessary condition for a local minimizer at an interior point. Hence no two distinct indices in the same time slice can both be interior. $\qquad \square$

### I.4. Core binarity for time-indexed TSP

**Theorem I.2** (Core binarity for time-indexed TSP). *Assume $c_{ij} \geq 0$ and fix $\varepsilon > 0$. If*

$$\gamma > C_{\max} + \varepsilon,$$

*then every local minimizer $x^\star$ of* (24) *satisfies $x_{i,t}^\star \in \{0,1\}$ for all $i, t$.*

*Proof.* Let $x^\star$ be a local minimizer of (24). Assume toward a contradiction that $J := J(x^\star) \neq \emptyset$, and pick an interior index $(i, t) \in J$.

By Lemma I.1, every variable in the same time slice $t$ other than $x_{i,t}^\star$ is binary, and every variable in the same city row $i$ other than $x_{i,t}^\star$ is binary. Hence $\sum_{k \neq i} x_{k,t}^\star \in \mathbb{Z}$ and $\sum_{s \neq t} x_{i,s}^\star \in \mathbb{Z}$. Write

$$\rho_t(x^\star) = \Big(\sum_{k \neq i} x_{k,t}^\star\Big) + x_{i,t}^\star - 1, \qquad \kappa_i(x^\star) = \Big(\sum_{s \neq t} x_{i,s}^\star\Big) + x_{i,t}^\star - 1.$$

Substituting these expressions into (28), the $x_{i,t}^\star$ terms cancel:

$$\frac{\partial V_{\text{time}}}{\partial x_{i,t}}(x^\star) = 2 \sum_{k \neq i} x_{k,t}^\star + 2 \sum_{s \neq t} x_{i,s}^\star - 2 \ \in \ 2\mathbb{Z}.$$

Since $x_{i,t}^\star$ is interior, first-order stationarity yields

$$0 = \frac{\partial \hat{f}_{\text{time}}}{\partial x_{i,t}}(x^\star) = \frac{\partial Q}{\partial x_{i,t}}(x^\star) + \varepsilon + \gamma \frac{\partial V_{\text{time}}}{\partial x_{i,t}}(x^\star).$$

By (25), $\frac{\partial Q}{\partial x_{i,t}}(x^\star) \geq 0$, hence the first two terms satisfy $\frac{\partial Q}{\partial x_{i,t}}(x^\star) + \varepsilon > 0$. Also by (26),

$$0 < \frac{\partial Q}{\partial x_{i,t}}(x^\star) + \varepsilon \leq C_{\max} + \varepsilon.$$

If $\frac{\partial V_{\text{time}}}{\partial x_{i,t}}(x^\star) \neq 0$, then since it is an even integer,

$$\left| \gamma \frac{\partial V_{\text{time}}}{\partial x_{i,t}}(x^\star) \right| \geq 2\gamma > 2(C_{\max} + \varepsilon) \geq \left| \frac{\partial Q}{\partial x_{i,t}}(x^\star) + \varepsilon \right|,$$

so the sum cannot be zero, a contradiction. Therefore $\frac{\partial V_{\text{time}}}{\partial x_{i,t}}(x^\star) = 0$, which forces $\frac{\partial Q}{\partial x_{i,t}}(x^\star) + \varepsilon = 0$, impossible because the left-hand side is strictly positive. Thus $J = \emptyset$ and $x^\star$ is binary. $\qquad \square$

### I.5. Feasibility of binary local minimizers

**Theorem I.3** (Feasibility of local minimizers for time-indexed TSP). *Assume $c_{ij} \geq 0$ and fix $\varepsilon > 0$. Suppose every local minimizer of* (24) *is binary. If*

$$\gamma > \tfrac{1}{2}(C_{\max} + \varepsilon),$$

*then every local minimizer $x^\star$ satisfies the assignment equalities $\sum_i x_{i,t}^\star = 1$ for all $t$ and $\sum_t x_{i,t}^\star = 1$ for all $i$. Consequently, $x^\star$ is feasible for* (22) *and encodes a Hamiltonian tour.*

*Proof.* Let $x^\star$ be a binary local minimizer. For a local minimizer over a box, every feasible direction $d$ in the tangent cone satisfies $\langle \nabla \hat{f}_{\text{time}}(x^\star), d \rangle \geq 0$.

**(A) No surplus rows/columns.** Suppose some time $t$ has $\sum_i x_{i,t}^\star \geq 2$, i.e. $\rho_t(x^\star) \geq 1$. Pick any $i$ with $x_{i,t}^\star = 1$ and take $d := -e_{i,t}$ (feasible since $x_{i,t}^\star = 1$). Using (28) and $\kappa_i(x^\star) \geq 0$ (city $i$ appears at least once when a 1 occurs), we have

$$\frac{\partial V_{\text{time}}}{\partial x_{i,t}}(x^\star) = 2\rho_t(x^\star) + 2\kappa_i(x^\star) - 4 \cdot 1 + 2 = 2\rho_t(x^\star) + 2\kappa_i(x^\star) - 2 \ \geq \ 0.$$

In fact, since $\rho_t(x^\star) \geq 1$, the right-hand side is at least 0 and is strictly positive unless $\kappa_i(x^\star) = 0$ and $\rho_t(x^\star) = 1$; either way, $\langle \nabla V_{\text{time}}(x^\star), d \rangle = -\partial_{i,t} V_{\text{time}}(x^\star) \leq 0$. Moreover, by (25), $\langle \nabla Q(x^\star), d \rangle = -\partial_{i,t} Q(x^\star) \leq 0$, and $\langle \nabla(\varepsilon \sum x)(x^\star), d \rangle = -\varepsilon < 0$. Therefore $\langle \nabla \hat{f}_{\text{time}}(x^\star), d \rangle < 0$, a contradiction. Hence every time $t$ satisfies $\sum_i x_{i,t}^\star \leq 1$. An identical argument shows every city $i$ satisfies $\sum_t x_{i,t}^\star \leq 1$.

**(B) No deficit rows/columns.** If some time $t$ had $\sum_i x_{i,t}^\star = 0$, then since every time has at most one 1 by (A), the total number of ones satisfies $\sum_{i,t} x_{i,t}^\star \leq n - 1$. Therefore some city $k$ must have $\sum_t x_{k,t}^\star = 0$ as well. Pick such a deficit pair $(k, t)$ with $x_{k,t}^\star = 0$ and take $d := +e_{k,t}$ (feasible). Then $\rho_t(x^\star) = \kappa_k(x^\star) = -1$, so (28) gives

$$\frac{\partial V_{\text{time}}}{\partial x_{k,t}}(x^\star) = 2(-1) + 2(-1) - 4 \cdot 0 + 2 = -2, \qquad \Rightarrow \qquad \langle \nabla V_{\text{time}}(x^\star), d \rangle = -2.$$

Also, by (26),

$$\langle \nabla Q(x^\star), d \rangle = \frac{\partial Q}{\partial x_{k,t}}(x^\star) \leq C_{\max}, \qquad \langle \nabla(\varepsilon \sum x)(x^\star), d \rangle = \varepsilon.$$

Therefore

$$\langle \nabla \hat{f}_{\text{time}}(x^\star), d \rangle \leq (C_{\max} + \varepsilon) - 2\gamma < 0,$$

contradicting local optimality when $\gamma > \frac{1}{2}(C_{\max} + \varepsilon)$. Hence no deficit time can exist; similarly no deficit city can exist.

Combining (A) and (B), every time has exactly one selected city and every city is selected at exactly one time, so $x^\star$ satisfies the assignment equalities and is feasible for (22). $\qquad\square$

*Remark* I.4 (Why the tie-break is harmless). On the feasible set of (15), we always have $\sum_{i,t} x_{i,t} = n$, so the added term $\varepsilon \sum_{i,t} x_{i,t}$ is constant and does not change which tour is optimal. It only simplifies exclusion of degenerate stationary points in the box relaxation.

