# OpenReview forum: "Local Minima in Quadratic-Penalty Relaxations of Binary Linear Programs"
_ICML.cc/2026/Conference — ICML 2026 regular_

### Official Review · Reviewer_ACcZ · 2026-02-16

**Soundness:** 1
**Presentation:** 1
**Significance:** 2
**Originality:** 3
**Overall Recommendation:** 3
**Confidence:** 3

**Summary:**

This paper studies relaxation of binary linear programming with quadratic penalties. The central motivation is to find feasible binary solutions by using the projected gradient method. To this end, the relaxation should not admit local minima that are fractional or infeasible. The paper provides a locally certifiable sufficient condition for ensuring that the relaxation has no such local minima, namely, the diagonal-free structure of penalties, integer-valued penalty gradients, and the local repairability. With this framework, the authors show that the standard relaxation for the open-pit mining problem may fail, and propose a modified relaxation that satisfies the sufficient condition. The authors further apply the framework to the 0-1 knapsack problem, traveling salesman problem, and their variants. Finally, experiments on those problems validate the theoretical findings.

**Compliance With Llm Reviewing Policy:**

Affirmed.

**Final Justification:**

The authors have addressed my questions and provided clarification regarding the direction set; however, the design of the penalty $V$ remains an issue, in general. Regarding scalability, the additional experiments in the rebuttal appear promising. Taking these points into account, I have updated my score from 2 to 3.

That said, there remain many issues to be addressed before this paper reaches an acceptable level, including improvements to writing quality and greater rigor in the treatment of numerical stability and complexity-theoretic aspects. For these reasons, my score remains a weak reject.

**Key Questions For Authors:**

How to define the direction set $D$ does not seem to be well explained in examples other than the open-pit mining problem. Is there any principled way to design $D$, or do we need to design it in an ad-hoc manner for each problem?

**Limitations:**

This is a minor issue, but the open-pit mining problem is solvable in polynomial time by reduction to the max flow problem. This should be clarified to avoid confusion.

**Strengths And Weaknesses:**

**Strengths:**

S1. The paper provides a principled framework for analyzing local minima of quadratic relaxations of binary linear programming. This can contribute to the advancement of gradient-based approaches for finding good feasible binary solutions to combinatorial optimization problems.

S2. The implication of the framework is interesting: for example, the standard relaxation for the open-pit mining problem may contain invalid local minima.


**Weaknesses:**

W1. The writing is not very clear. Some parts are problematic and make it difficult to properly assess the technical contributions. For example:
1. Lines 132--133: texts are overlapping. This appears to be caused by excessive negative vertical spacing and should be fixed.
2. Line 114: "objective weights" probably refer to $w$, but should be clarified.
3. Theorem 2.5: $c_{iq}$ is undefined, and it does not seem to be used elsewhere.
4. Definition 2.6: What "assignments" refer to is not clear.
5. Section 2.2: "binary-equivalent" is used before it is defined in Definition 3.4.
6. Section 3.2.2: $m_k$ is undefined.

W2. In Section 3.2.2, the slack variables include $2^\ell$, which may significantly increase the bit-length of the problem and thus should be treated with care.

W3. The above issue may also harm numerical stability in practice, as the problem size grows. The instance sizes used in the experiments do not seem to be clarified, and so it is unclear whether the proposed relaxation is scalable to large instances in terms of numerical stability.

---

> ### Author Rebuttal · Authors · 2026-03-31
>
> We thank the reviewer for the careful reading. We will improve the pesentation revise the paper accordingly. We will clarify the notation and improve the exposition to make the arguments easier to follow. We address the concerns below.
>
> # (W1) Clarity, notation, and organization
> We will fix the formatting issues, clarify notation and terminology, move the definition of “binary-equivalent” earlier, and revise the presentation around Definitions 2.2–2.6 so that the roles of core variables, feasible assignments, and repair directions are explicit before they are used in the examples. We believe these revisions will make the presentation clearer and easier to follow, without affecting the technical content or the validity of the results.
>
> # (Q1) Direction set $D$: principled design rather than ad hoc choice
>
> Our intention is that $D$ be **problem-specific but principled**, not ad hoc. It refers to the family of bounded local repair moves used to certify condition (C3), which identify local certificates of infeasibility, select simple bounded edits that reduce the violation while respecting the box geometry, and guarantee that at least one such edit yields a uniform penalty decrease.
>
> In open-pit mining, ancestor forms are formulated such that single-block removals lead to a decrease in the penalty. Likewise, for knapsack problems, a feasible direction set is $D = \\{ -e_i: i = 1,\dots,n \\}$ which turns off one active item/vertex participating in a violated constraint at an infeasible binary point. We will add the explicit definition of $D$ for each example.
>
> # (W2) + (W3) Slack bits, numerical stability, and scalability
> We will include a more detailed discussion of the complexity of slack encoding in the revision. In particular, each inequality requires only $\mathcal{O}(\log b_k)$ slack variables, which keeps the dimensional overhead modest. However, we acknowledge that such encodings may worsen the conditioning of the resulting optimization problem, necessitating preconditioning techniques (e.g., Adam) in practice. In the revision, we will explicitly report instance sizes, runtime, and iteration statistics, and clarify that this issue should be viewed as a practical limitation of the current construction rather than a limitation of the underlying theory.
>
> To better support the motivation of scalability, we also added a fixed-time-budget maximum independent set experiment (packing setting), in which PGD outperformed Gurobi on all 24 tested ER graph settings. In addition, we show that the leading heuristic ReduMIS can still struggle to produce prompt solutions on larger/denser instances, supporting the need to study the QUBO formulation. We refer the reviewer to Table A and Table B of https://anonymous.4open.science/r/Quadratic-Penalty-Relaxations-EC11/rebuttal_table.md.
>
> # (L1) Open-pit mining as an example
> The open-pit mining problem, in its simplest form, is polynomial-time solvable. We use it primarily as a theoretical example to illustrate how our theorems can guide the choice of QUBO formulations with local correctness. This type of guidance can be extended to more complex settings, such as open-pit mining with additional linear constraints, which becomes NP-hard.
>
> We appreciate the reviewer’s emphasis on clarity and reproducibility. In the revision, we will improve the exposition, make the role of $D$ explicit as a principled repair family, and better report practical scaling information so that the paper’s theory and examples are easier to assess and build upon.

---

> > ### Author Rebuttal · Reviewer_ACcZ · 2026-04-01
> >
> > I appreciate the authors' response.
> >
> > **Q1.** I understand that the direction set is designed with C3 in mind. However, since its construction is necessarily problem-specific, it remains unclear what justifies the claim that the approach is principled. For instance, in the case of TSP, through what natural line of reasoning would one arrive at a particular choice of $D$?
> >
> > **W2.** My concern regarding $2^\ell$ also has a complexity-theoretic aspect. Just as the standard DP for the knapsack problem is treated as a pseudo-polynomial-time algorithm (because it depends explicitly on a value that is exponential in the bit length $\ell$), methods that explicitly rely on quantities of the form $2^\ell$ warrant careful remark on their computational complexity, as well as numerical stability. I encourage the authors to discuss this point in the paper.

---

> > > ### Author Response · Authors · 2026-04-02
> > >
> > > ### (Q1)
> > > Thank you for the follow-up question. We clarify the roles of $V$ and $D$ in our framework.
> > >
> > > The penalty $V$ is designed in a problem-dependent manner, but this design is guided by (towards satisfying) the common structural conditions (C1)–(C3). In contrast, once $V$ is fixed and satisfies (C3), the direction set $D$ can be chosen in a principled and canonical way.
> > >
> > > Indeed, suppose $z$ is an infeasible point. By (C3), there exists a direction $d \in T(z)$ （$T(z)$ defined in Def 2.2） such that $\langle \nabla V(z), d \rangle \le -\delta$ for some $\delta > 0$. Write $d = \sum_{i=1}^n c_i e_i$. Then
> > > $\langle \nabla V(z), d \rangle = \sum_{i=1}^n c_i \langle \nabla V(z), e_i \rangle$.
> > > Since the sum is negative, there exists an index $i$ such that
> > > $c_i \langle \nabla V(z), e_i \rangle < 0$.
> > >
> > > Let $\tilde d_i = \mathrm{sign}(c_i) e_i$. Then
> > > $\langle \nabla V(z), \tilde d_i \rangle = \mathrm{sign}(c_i)\langle \nabla V(z), e_i \rangle < 0$.
> > >
> > > Moreover, since $d \in T(z)$, we have
> > > $z_i = 0 \Rightarrow c_i \ge 0$,
> > > $z_i = 1 \Rightarrow c_i \le 0$,
> > > $0 < z_i < 1 \Rightarrow c_i \in \mathbb{R}$,
> > > which implies $\tilde d_i \in T(z)$. Hence $\tilde d_i =sign(c_i)e_i$ is a also feasible descent direction.
> > >
> > > Therefore, it is sufficient to take $D = \\{\pm e_i : i = 1, \dots, n\\}$ or $D = \\{\pm e_i : i = 1, \dots, n\\} \cap T$, showing that once $V$ satisfies (C3), the choice of $D$ is canonical and does not depend on the specific problem.
> > >
> > > If the question is instead about how to construct a suitable penalty $V$, we agree that this is currently problem-dependent and does not admit a fully general procedure. However, the design is guided by the structural conditions (C1)–(C3), which provide concrete criteria rather than ad hoc choices.
> > >
> > > For relatively simple problems, such as the assignment problem with least-squares penalties, after making it diagonal free, (C3) can be verified directly. For other problems, additional design effort is required to ensure local repairability. There may also be problem formulations for which no penalty $V$ satisfying (C1)–(C3) can be identified within our framework.
> > >
> > > ### (W2)
> > > We thank the reviewer for the clarification. We will make this discussion explicit in the revision.

---

### Official Review · Reviewer_pUpH · 2026-02-28

**Soundness:** 3
**Presentation:** 3
**Significance:** 3
**Originality:** 3
**Overall Recommendation:** 4
**Confidence:** 2

**Summary:**

This paper addresses a fundamental challenge in using gradient-based optimization for combinatorial problems: the tendency of continuous relaxations to converge to fractional or infeasible local minima. The authors propose a theoretical framework for Quadratic Unconstrained Binary Optimization (QUBO) problems relaxed over the hypercube $[0, 1]^n$.

The core contribution is the identification of three sufficient conditions—Diagonal-free structure (C1), Integer-valued gradients (C2), and Local repairability (C3)—which guarantee that for a sufficiently large (but finite) penalty parameter $\gamma$, every local minimum of the relaxed objective is both binary and feasible. The authors demonstrate the utility of this theory by redesigning penalty terms for Open-Pit Mining, 0-1 Knapsack, and Traveling Salesman Problems (TSP).

**Compliance With Llm Reviewing Policy:**

Affirmed.

**Key Questions For Authors:**

1.  **Complexity Scaling:** For the Open-Pit Mining problem, how does the number of terms in the "Ancestor" formulation scale relative to the "Parent" formulation as the graph depth increases? Does this lead to a bottleneck in wall-clock time for large-scale instances?
2.  **Optimizer Robustness:** The theory is built around PGD local minima. To what extent do these guarantees hold when using adaptive optimizers like Adam, which were used in your Knapsack experiments to handle ill-conditioning?
3.  **Basins of Attraction:** Does the theory-guided formulation increase the size of the basins of attraction for "good" feasible solutions, or does it primarily focus on eliminating "bad" fractional ones?

**Limitations:**

See weaknesses.

**Strengths And Weaknesses:**

**Strengths**
*   **Theoretical Contributions:** The paper provides a formal characterization of "spurious" local minima and proves that traditional penalty approaches often fail because they create "flat" regions at infeasible vertices. The "Local Repairability" condition is a novel and rigorous solution to this problem.
*   **Practical Framework:** Unlike many heuristic-based penalty methods, this work offers a constructive way to design differentiable proxies. The derivation of finite lower bounds for the penalty parameter $\gamma$ is highly valuable for practitioners.
*   **Empirical Validation:** The comparison between "Naive" and "Theory-guided" formulations (specifically the Ancestor vs. Parent formulations in mining) clearly illustrates how structural properties of the penalty term directly impact the success of first-order optimizers.


**Weaknesses**
*   **Gap Between Local and Global Optima:** While the theory guarantees that local minima are *feasible* and *binary*, it does not guarantee their *quality*. In NP-hard problems, a feasible local minimum may still be significantly worse than the global optimum.
*   **Complexity of Formulations:** The "Theory-guided" formulations often require a significantly larger number of penalty terms (e.g., the Ancestor formulation in Open-Pit Mining). The paper lacks a detailed analysis of the trade-off between the theoretical correctness of the landscape and the increased computational cost (memory and time) per gradient step.
*   **Optimization Difficulty:** High penalty values ($\gamma$), while theoretically sound, can lead to ill-conditioning. This makes the objective landscape "sharp," which may hinder the convergence of standard gradient descent algorithms like PGD.

---

> ### Author Rebuttal · Authors · 2026-03-31
>
> We thank the reviewer for the thoughtful review and for recognizing the main contribution as a local-correctness theory together with a constructive formulation-design principle.
> # (W1) Gap between local and global optima
> We agree that our theory is local, not global. The guarantee is that, under (C1)–(C3) and sufficiently large $\gamma$, every PGD local minimum is binary and feasible. It does **not** claim global optimality or near-optimality. We will add this clarification in the revision.
> # (W2) + (Q1) Complexity scaling / trade-off in the ancestor formulation
> We agree that the theoretical correctness of the ancestor penalty comes at the cost of increased complexity. In this work, we demonstrate that the ancestor formulation is a safe construction that satisfies the sufficient conditions established in the paper. At the same time, we do not rule out the possibility that alternative penalty constructions may exist with lower complexity while retaining the desired guarantees.
>
> The complexity of the **ancestor** and **parent** formulations is $\mathcal{O}(n+A)$ and $\mathcal{O}(n+m)$ respectively, where $m$ and $A$ denote the number of edges in the original precedence DAG and the number of edges in the transitive closure of that DAG. We also added a numerical comparison of their runtimes on two real-world instances from MineLib, `newman1` and `kd`.
>
> newman1: 1,060 blocks and 3,922 precedence arcs.
>
> kd: 14,153 blocks and 219,778 precedence arcs.
> ### Table C. Ancestor VS Parent Formulation for Open-Pit Mining with $200$ Runs for Each Setting.
> | instance | mode | mean_steps | median_steps | min_steps | max_steps | mean_time_sec | median_time_sec | min_time_sec | max_time_sec |
> | ---      | ---  |  ---         | ---          | ---       | ---       | ---           | ---             | ---          | ---          |
> | newman1 | ancestor | 1092.165 | 1091 | 693 | 1557 | 0.988636 | 0.97643 | 0.560832 | 1.884777 |
> | newman1 | parent | 1018.605 | 967 | 477 | 1846 | 0.747483 | 0.712712 | 0.340681 | 2.177611 |
> | kd | ancestor | 537.72 | 534 | 498 | 632 | 0.968356 | 0.959496 | 0.879048 | 1.833711 |
> | kd | parent | 517.77 | 516 | 462 | 644 | 0.443613 | 0.434604 | 0.377893 | 0.797381 |
>
> These results confirm that the ancestor formulation is more expensive to solve. In the tested real instances, this cost remains affordable while delivering the desired local-correctness behavior.
>
> # (Q2) + (W3) Optimizer robustness / large penalties
> We use projected Adam in knapsack instances with slack variables to mitigate ill-conditioning and anisotropy. Adam alleviates this issue via diagonal preconditioning. Since the preconditioning factors are strictly positive, the set of first-order stationary points for our problem remains unchanged. While the momentum term in Adam may alter the optimization trajectory and induce mild oscillations around a local minimizer, it does not affect the first-order optimality conditions. In our experiments, such oscillations are negligible under the chosen momentum parameters.  One can reduce the strength of momentum to avoid oscillation when necessary. We will add a discussion to make this separation clearer: the theory is for PGD-local correctness, while Adam is used as a practical robustness choice in harder instances.
>
> In addition, the large threshold appears to be primarily due to the ill-posedness of the dataset. Specifically, the MineLib instances contain very large-magnitude values, which drive the objective weights to scales on the order of 1e16. Consequently, $γ$ is set at a comparable scale. In the revision, we will incorporate data preprocessing steps to normalize or remove these values, following approaches proposed in some prior studies.
>
> # (Q3) Basins of attraction
> Our current theory focuses only on the **elimination of undesirable (fractional or infeasible) local minima**. It does not provide guarantees for identifying flatter minimizers. We agree that this is an interesting direction for future work.
>
> We appreciate the reviewer’s framing of the paper as a constructive theory for designing reliable relaxed objectives. In the revision, we will sharpen the local-vs-global scope, make the formulation-size trade-off explicit, and include the open-pit runtime comparison above to show that the price of the safe ancestor construction is affordable on demonstrated real instances.

---

> > ### Author Rebuttal · Reviewer_pUpH · 2026-04-03
> >
> > I thank the authors for the rebuttal. My specific concerns regarding complexity scaling (Table C) and the theoretical justification for using Adam are resolved.
> >
> > However, given my lower confidence (2) in empirical benchmarking, I note Reviewer pgpZ's valid concerns about the chosen baselines in the newly added experiments. I defer to Reviewer pgpZ's expertise on the empirical evaluation and will maintain my "Weak Accept" score based solely on the paper's theoretical contributions.

---

> > > ### Author Response · Authors · 2026-04-07
> > >
> > > We appreciate the reviewer’s recognition of the theoretical contribution of our work, which is indeed the main focus of the paper.
> > >
> > > Gradient-based optimization over relaxed QUBO formulations is already used in practice (e.g., as scalable local search or refinement procedures), yet lacks a clear theoretical understanding of when it produces valid binary and feasible solutions. This gap serves as the main motivation for our work.
> > >
> > > The additional experiments included in the rebuttal are intended to further support this motivation and provide additional context, such as to demonstrate that existing relaxed QUBO formulations can achieve competitive performance compared to modern baselines in certain settings.
> > >
> > > The core contribution of the paper remains the structural understanding of when such methods can be reliably applied. We will make this motivation and scope more explicit in the revision.

---

### Official Review · Reviewer_Z9xA · 2026-03-04

**Soundness:** 3
**Presentation:** 2
**Significance:** 3
**Originality:** 2
**Overall Recommendation:** 4
**Confidence:** 4

**Summary:**

This paper studies when projected gradient descent (PGD) on box-relaxed QUBOs can be trusted to return binary and feasible solutions. The authors show that some standard quadratic penalties can have infeasible binary points that are first-order stationary under the box constraints and may remain strict local minima even as the penalty weight grows. They then give a general sufficient criterion for “local correctness,” based on diagonal-free penalties on core variables, integer-valued structure, and a local repairability condition that ensures a penalty-decreasing feasible direction from any infeasible binary core. Under these conditions, they prove an explicit finite threshold on the penalty weight so that every PGD local minimum is binary on the core and feasible. They apply the framework to open-pit mining, knapsack, and TSP, and experiments show the theory-guided formulations achieve consistent feasibility while naive QUBOs can fail.

**Compliance With Llm Reviewing Policy:**

Affirmed.

**Key Questions For Authors:**

1. The multiconstraint knapsack section uses an overcorrection with $\alpha \geq 2$ (you take $\alpha=2$). Later, the MTZ-TSP construction uses a related overcorrection in the appendix. Could you clearly define or explain $\alpha$ in the main text?
2. For knapsack, the penalty uses binary slack expansions, and you mention the landscape can become ill-conditioned (so you use projected Adam). How does the number of slack bits scale with the capacities $b_k$? In practice, does this limit the size of inequality problems that your approach can handle?
3. Do you have basic runtime (running time) results for PGD/Adam on your theory-guided QUBOs, at least on moderate-size instances? If not, could you clarify whether the main goal is a competitive solver, or mainly a correctness theory and penalty design guide?
4. Some thresholds $\gamma^*$ reported in the experiments are extremely large (e.g., open-pit). When  $\gamma$ is very large, the objective can be badly scaled. How sensitive are PGD/Adam and automatic differentiation to this scaling?

**Limitations:**

yes

**Strengths And Weaknesses:**

Strengths:
1. The paper shows that even with very large penalty weights, a box-relaxed QUBO can still have infeasible binary points that are strict local minima under the PGD first-order condition. This explains why “just increase $\gamma$” may not fix feasibility in continuous relaxations.
2. The authors give a clean, reusable criterion (diagonal-free core, integer structure, and a local repairability condition) and prove a finite penalty threshold so that every PGD local minimum is binary (on the core) and feasible. They also apply the same ideas to open-pit mining, knapsack, and TSP, and show that theory-guided penalties avoid the failures seen in naive QUBOs.

Weaknesses:
1. The experiments are small: only two instances per problem class and a few random starts (Tables 1-3). This is not enough to show how the approach scales. It would help to test more and larger instances, and to report basic runtime or iteration counts, not only binarity/feasibility rates.
2. The paper uses an overcorrection with $\alpha=2$ in the knapsack and MTZ-TSP constructions. The appendix explains why $\alpha=1$ can fail (e.g., Lemma H.1 for MTZ), but the main text does not clearly introduce $\alpha$ and explain why choosing 2 is natural, and whether other values (e.g., $\alpha>2$) would also work.
3. For inequality constraints, the method introduces slack variables via binary expansion, which increases the dimension. The paper also notes this can make the landscape ill-conditioned, and uses projected Adam for knapsack instead of PGD. This gap between the PGD-based theory and the optimizer used in experiments should be discussed more clearly, since it may limit real-world scalability.

---

> ### Author Rebuttal · Authors · 2026-03-31
>
> We thank the reviewer for highlighting both the reusable theoretical criterion and the practical questions around overcorrection, slack expansion, optimizer choice, and scaling.
>
> # (W1) + (Q3) Runtime / scalability / intended role of the method
> We agree that the current experiments are small and primarily theory-validating. To better support the scalability motivation, we added a fixed-time-budget experiment on **Maximum Independent Set (MIS)**, as a special packing problem mentioned in the paper. On Erdős–Rényi graphs with graph size $n$ ranging from $2000$ to $40000$ and edge density $p \in \\{0.3,0.5,0.7\\}$, 8 instances per $(n,p)$ setting were tested, and PGD returned a larger independent set than Gurobi under the same **5-minute** budget on **all 24 tested settings**.
>
> ### Table A. Average MIS Solution Size of PGD and Gurobi Under 5 Minutes Time Budget on 8 Erdős–Rényi Instances.
> | Optimizer|PGD|PGD |PGD|Gurobi|Gurobi|Gurobi|
> | --- | --- | --- | --- | --- | --- | --- |
> | (n,p) | 0.3 |0.5 | 0.7 |  0.3 | 0.5 | 0.7 |
> | 2000 | 27 | 14.25 | 10 | 21.875 | 13.125 | 8.375 |
> | 4000 | 28.5 | 16 | 10.5 | 20.5 | 11.625 | 7.375 |
> | 6000 | 30.25 | 17.375 | 10.5 | 21.75 | 12.25 | 8 |
> | 8000 | 30.875 | 17.375 | 10.75 | 23.25 | 13.25 | 7.75 |
> | 10000 | 30.625 | 17 | 10.625 | 22.875 | 13.5 | 8.25 |
> | 20000 | 32.5 | 17.875 | 11 | 24.5 | 14 | 8.625 |
> | 30000 | 33.625 | 18.75 | 11.25 | 26.25 | 14.25 | 9.125 |
> | 40000 | 34.125 | 18.875 | 11.625 | 26.75 | 15.125 | 9.625 |
>
> We also ran ReduMIS, the best known heuristic for solving the MIS problem. For the tested settings, the **average** time to return the first solution already exceeds the 5-minute limit for every completed case, which reinforces the insight that gradient-based local solvers are attractive in fixed-time regimes, particularly for larger or denser packing instances.
>
> ### Table B. ReduMIS Preliminary Time-to-First-Solution Time in Seconds.
> | (n,p)     | 0.3    | 0.5    | 0.7    |
> |-------|-----------:|-----------:|-----------:|
> | 2000  | 309.008750 | 327.442750 | 368.753375 |
> | 4000  | 346.855500 | 608.526875 | 820.273750 |
> | 6000  | 474.935250 | 994.785375 | 1258.280875 |
> | 8000  | 830.066125 | 1578.481000 | — |
> | 10000 | 783.218250 | 1625.062625 | — |
> | 20000 | 2510.655000 | 2407.158750 | — |
> | 30000 | 3533.131250 | 4333.006667 | —
>
> We will present this experiment as supplementary evidence supporting the need to study the QUBO formulation.
>
> # (W3) + (Q2) Slack expansions, optimizer choice, and practical scaling
> For each inequality $k$, representing slack values up to the RHS $b_k$ requires $m_k+1=\lceil \log_2(b_k+1)\rceil$ bits, so the dimensional increase is **logarithmic** in the RHS/capacity, not linear in $b_k$. Although the number of slack variables is only $\lceil \log_2(b_k+1)\rceil$, they could still make the problem more ill-conditioned.
>
> In practice, we additionally use projected Adam in knapsack instances with slack variables to mitigate ill-conditioning. Adam alleviates this issue via diagonal preconditioning. Since the preconditioning factors are strictly positive, the set of first-order stationary points for our problem remains unchanged. While the momentum term in Adam may alter the optimization trajectory and induce mild oscillations around a local minimizer, it does not affect the first-order optimality conditions. In our experiments, such oscillations are negligible under the chosen momentum parameters.  One can reduce the strength of momentum to avoid oscillation when necessary. We will add a discussion to make this separation clearer: the theory is for PGD-local correctness, while Adam is used as a practical robustness choice in harder instances.
>
> # (W2) + (Q1) Overcorrection parameter $\alpha$
> Thank you for your comments. In the knapsack construction, $\alpha=1$ gives exact binary equivalence, but it may fail the local-repairability property required by our theory. We use $\alpha=2$ because it is the **smallest overcorrection** that restores the needed local descent structure. Larger values may also work, but are not needed for our results. We will make this point clear in the revision.
>
> # (Q4) Very large $\gamma$ and conditioning
> The large threshold appears to be primarily due to the ill-posedness of the dataset. Specifically, the MineLib instances contain very large-magnitude values, which drive the objective weights to scales on the order of $1e16$. Consequently, $γ$ is set at a comparable scale. In the revision, we will incorporate data preprocessing steps to normalize or remove these values, following approaches proposed in some prior studies.
>
>
> We appreciate the reviewer’s focus on the practical side of the framework. In the revision, we will make the role of $\alpha$, slack scaling, and the PGD/Adam distinction much clearer, and we will add the fixed-budget MIS evidence to better support the scalability motivation while keeping the paper’s main contribution on the theory side.

---

> > ### Author Rebuttal · Reviewer_Z9xA · 2026-04-01
> >
> > My concerns have been adequately addressed

---

> > > ### Author Response · Authors · 2026-04-08
> > >
> > > We sincerely thank the reviewer for the encouraging feedback and for indicating that the concerns have been fully resolved.

---

### Official Review · Reviewer_pgpZ · 2026-03-10

**Soundness:** 3
**Presentation:** 2
**Significance:** 3
**Originality:** 3
**Overall Recommendation:** 4
**Confidence:** 4

**Summary:**

The article studies the properties of penalty terms in QUBO formulations of combinatorial optimization problems that enable the use of gradient-based methods to obtain feasible solutions. In particular, it is shown that a diagonal-free penalty term structure, integer-valued penalty gradients, and local repairability are sufficient to ensure that all local minima of the box relaxation are integral and do not violate the penalty term. New QUBO formulations that fulfill these properties are presented for the open-pit mining, the 0-1 knapsack and the travelling salesman problem. These formulations are compared experimentally with existing ones on two instances per problem, validating the theoretical results.

**Compliance With Llm Reviewing Policy:**

Affirmed.

**Final Justification:**

I thank the authors for their response and the clarifications provided. I think that the theoretical results are interesting, and I have increased my score accordingly. At the same time, I still have some concerns about the experiments conducted and whether the writing can be improved for the camera-ready version of the article.

**Key Questions For Authors:**

- How do the integer feasible solutions found by the proposed method compare in terms of objective value and computation time to those found by other efficient algorithms for the respective problems?
- From my understanding, Theorem 2.8 implies that, for decision problems, finding a formulation satisfying the given conditions would imply them to be efficiently solvable. Is this correct? Can this statement be applied to prove that certain decision problems can be solved efficiently?

**Limitations:**

yes

**Strengths And Weaknesses:**

The theoretical results are significant, and I believe them to be correct and new. Although I have not had the time to check all details, I have given the main arguments some scrutiny, and I have not found a mistake. At the same time, the presentation and discussion need to be improved, especially regarding the following two aspects:

I find it difficult to understand why the integrality property of Theorem 2.8 does not imply that the optimal solutions to the LP relaxation of the original problem are integral and can therefore be found efficiently. I do understand that this is because the penalty terms are non-zero for the fractional feasible solutions, which makes their objective value worse than that of the optimal integral feasible solution. Still, this should be discussed more explicitly, to avoid confusion. Moreover, the upper bound on the optimal objective value (in the case of minimization) that results from this should be discussed.

In the introduction, the scalability of gradient-based methods for solving combinatorial optimization problems is mentioned as a major motivation. The experiments, however, do not align with this motivation. Only two instances are considered per problem, and only the fraction of integral and feasible solutions are reported, which are guaranteed by the theoretical results. The article would benefit greatly from a comparison of the solution quality and runtimes obtained by the presented method with those obtained by existing heuristic solvers on suitable benchmark instances.

In addition to these major aspects, the following minor points should be addressed:
- Both the abstract and the introduction mention ensuring the feasibility for the QUBO problem. I find this statement confusing (because QUBO is unconstrained) and suggest rephrasing it.
- Line 85, Left: "correct" -> "feasible"
- Line 67, Right: I suggest making related studies its own section and including work on other heuristic solvers.
- Line 80, Right: "The paper in ..." seems to be wrong.
- Line 106, Right: I suggest introducing QUBO in its standard form and then stating that we can obtain (1) by introducing auxiliary variables.
- Lines 148-156, Left: By Definition 2.3, $V(z)$ does not have $z_i^2$ terms. I suggest either changing $i<j$ to $i\leqj$ here, or adapting Section 2.2 accordingly.
- Lines 158-159, Left: I do not understand why the interaction graph should only be defined with respect to the square-free part of the penalty for core variables. Is this formulation correct?
- Lines 114, Right: $V(x)$ is not a matrix, so the term "diagonal-free" does not apply here. I suggest rephrasing this.
- Line 115, Right: In Lemma 2.4 and in the remainder of the article, the term "local minimizer" is used for the "local minimum of PGD" from Definition 2.2. I suggest clarifying this.
- Lines 119-121, Right: In Lemma 2.4, quantifiers are missing.
- Line 134, Right: In Theorem 2.5, the coefficients $c_{iq}$ should be $Q_{ij}$.
- Line 152, Right: In Definition 2.6, $\mathcal{C}$ is no longer defined.
- Line 165, Left: "For sufficiently large ..." -> "For any ...".
- Lines 172-173, Right: Since $p$ is never used in the following, I suggest removing the following line: "... for maximization of linear profits $p$, we set $w := −p$".
- Line 225, Left: References seem to refer to the wrong sections here, without links.
- Line 257, Left: The standard unit vectors $e_i$ are not defined.
- Lines 224-245, Right: From my understanding, the case described in Remark 3.2 also holds for infinite penalties, in accordance with the claims in the introduction. I thus suggest removing the following line: "Why the naive parent formulation can require “infinite” penalty".
- Lines 267-274, Right: This paragraph seems to be misplaced, since the open-pit mining problem is considered previously.
- Line 428, Right: "correct" -> "feasible"

---

> ### Author Rebuttal · Authors · 2026-03-31
>
> We thank the reviewer for the careful reading and for recognizing the theoretical contribution as significant and new.
>
> # (W1) + (Q2) Scope of Theorem 2.8: LP relaxation and efficient solvability
> Theorem 2.8 only applies to **quadratic penalized box relaxation**, which does not include the ordinary LP relaxation as a special case. So our result does not imply LP-relaxation integrality, and LP-relaxation feasible solutions could still be fractional as you correctly mentioned. We will add a discussion of the LP-relaxation in the revision.
>
> # (W2) + (Q1) Scalability / runtime / comparison with other solvers
> Thank you for the comment. We added a large-scale experiment on the **maximum independent set (MIS)** discussed in the paper as a special canonical packing problem. The experiment is carried out under a fixed time limit of 5 minutes. We evaluate graphs across different sizes (given by  $n$) and densities (given by $p$). In particular, we used Erdős–Rényi graphs with $n \in \\{2000,4000,6000,8000,10000,20000,30000,40000\\}$, $p \in \\{0.3,0.5,0.7\\}$. We tested 8 instances for each $(n,p)$ setting and compared PGD against Gurobi under the same **5-minute** budget. The largest graph in this experiment contains 560 million edges.
>
> We observe that PGD consistently returns a larger independent set than Gurobi:
>
> ### Table A. Average MIS Solution Size of PGD and Gurobi Under 5 Minutes Time Budget on 8 Erdős–Rényi Instances.
> | $(n, p)$ | PGD (0.3) | Gurobi (0.3) | PGD (0.5) | Gurobi (0.5) | PGD (0.7) | Gurobi (0.7) |
> |---|---:|---:|---:|---:|---:|---:|
> | 2000  | 27     | 21.875 | 14.25  | 13.125 | 10     | 8.375 |
> | 4000  | 28.5   | 20.5   | 16     | 11.625 | 10.5   | 7.375 |
> | 6000  | 30.25  | 21.75  | 17.375 | 12.25  | 10.5   | 8     |
> | 8000  | 30.875 | 23.25  | 17.375 | 13.25  | 10.75  | 7.75  |
> | 10000 | 30.625 | 22.875 | 17     | 13.5   | 10.625 | 8.25  |
> | 20000 | 32.5   | 24.5   | 17.875 | 14     | 11     | 8.625 |
> | 30000 | 33.625 | 26.25  | 18.75  | 14.25  | 11.25  | 9.125 |
> | 40000 | 34.125 | 26.75  | 18.875 | 15.125 | 11.625 | 9.625 |
>
> We also ran **ReduMIS**, a leading heuristic for the MIS problem. As shown in the table below, the **average** time for ReduMIS to return its first solution already exceeds the 5-minute budget (ranging from **309s** on $ER (2000, 0.3)$ to **4333s** on $ER (30000, 0.5)$) in the current settings. We therefore use these numbers only as supporting evidence showing that even strong heuristic methods can still struggle to produce prompt solutions on larger/denser instances.
>
> ### Table B. ReduMIS Preliminary Time-to-First-Solution Time in Seconds.
> | (n,p)     | 0.3    | 0.5    | 0.7    |
> |-------|-----------:|-----------:|-----------:|
> | 2000  | 309.008750 | 327.442750 | 368.753375 |
> | 4000  | 346.855500 | 608.526875 | 820.273750 |
> | 6000  | 474.935250 | 994.785375 | 1258.280875 |
> | 8000  | 830.066125 | 1578.481000 | — |
> | 10000 | 783.218250 | 1625.062625 | — |
> | 20000 | 2510.655000 | 2407.158750 | — |
> | 30000 | 3533.131250 | 4333.006667 | — |
>
> We will present this experiment as supplementary evidence supporting the need to study the QUBO formulation.
>
> # (M1) Minor presentation / notation issues
> Thank you for the detailed writing suggestions. We will revise the terminology around “feasibility”, clarify the distinction between local minima and PGD-local points, fix the missing definitions / quantifiers / notation issues, and improve the organization of the related-work and examples sections.
>
> We appreciate the reviewer’s emphasis on both conceptual scope and empirical motivation. In the revision, we will make the LP/complexity scope precise, add the fixed-budget MIS experiment above, and improve the presentation accordingly. Our goal is to state the contribution clearly as a **theory of when relaxed quadratic penalties are locally trustworthy for gradient-based search**, supported by targeted empirical evidence.

---

> > ### Author Rebuttal · Reviewer_pgpZ · 2026-04-02
> >
> > I appreciate the author's response, however, some of my questions and concerns have not been adequately addressed, as outlined below.
> >
> > **W1, Q2:** I am aware that Theorem 2.8 only applies to quadratic penalized box relaxations. However, my remark and my question regard the implications of this for the original problems. Consider for example the maximum independent set problem formulated as a binary linear program. On the one hand, given an instance of this problem, we can solve the LP relaxation efficiently and obtain a potentially fractional solution $x^*$. On the other hand, as shown by the authors, we can also construct an equivalent instance of the QUBO problem with penalty terms that satisfies the conditions of Theorem 2.8.
> >
> > From my understanding, Theorem 2.8 then implies that the solution to the QUBO problem corresponding to $x*$ is not a local minimum of the box relaxation. I remarked that I initially found it difficult to understand how $x^*$ could be a local minimum of the LP relaxation but not of the box relaxation, and suggested that this should be made clearer. My question regards the case where the initial problem is a binary decision problem, i.e. when we only want to determine whether a binary solution that satisfies a given set of linear constraints exists. For this kind of problem, obtaining a QUBO formulation satisfying the conditions of Theorem 2.8 would, from my understanding, allow to solve the original problem efficiently. Is this understanding correct?
> >
> > **W2, Q1:** The quality of the solutions obtained by PGD is convincing. However, I have significant reservations regarding the baselines selected for the experiments. Due to the size of the instances and the time limit imposed, neither Gurobi nor ReduMIS are suitable baselines, and comparing them with PGD exaggerates the performance of the proposed method. Instead, I would strongly recommend that the authors consider using more efficient heuristics designed for such large instances. For example, I implemented a simple greedy degree algorithm that iteratively adds the vertex with the lowest degree to the independent set, and achieved a mean size of $10.75$ in $31.79$ seconds for $n=40000$, $p=0.7$ on eight instances. A simple stochastic local search algorithm achieved even a mean size of $12.25$ in $32.05$ seconds on the same instances.

---

> > > ### Author Response · Authors · 2026-04-05
> > >
> > > ## W1, Q2
> > > We thank the reviewer for the clarification. The key difference between our formulation and the standard LP relaxation is that our construction of $V$ enforces binarity at local minimizers, whereas the LP relaxation only guarantees feasibility. Consequently, LP solutions may be fractional, while our formulation is designed so that local minimizers correspond to binary feasible solutions, which is a smaller set.
> > >
> > > For the binary decision problem, yes, obtaining a QUBO formulation satisfying the conditions of Theorem 2.8 would allow the problem to be solved efficiently using PGD in the following sense: the standard convergence guarantee of PGD for nonconvex smooth problems over a box-constrained domain is that it converges to a first-order stationary point at a rate $\mathcal{O}(1/T)$.
> > >
> > > To further avoid convergence to strict saddle points, it is well known that introducing small perturbations (e.g., noise injection or stochasticity) enables the iterates to escape saddle regions with high probability, so that the method converges to local minimizers almost surely.
> > >
> > > ## W2, Q1
> > >
> > > Our original choice of baselines essentially follows recent literature on MIS solvers, e.g., differential meta-solver [DIMES], diffusion-based method [DIFFUSCO], GNN-based methods [OptGNN,CRA], and reinforcement learning method [LwD]. We found in all of these studies, comparisons are typically made against a commercialized ILP solver such as Gurobi, and the SOTA heuristics, KaMIS (aka ReduMIS).
> > >
> > > Following the reviewer’s suggestion, we have added comparisons with simpler heuristics that can return a solution faster, including stochastic local search (SLS) (stochastic perturbation with ratio 25% + greedy repair) and a simplified version of GRASP (Semi-greedy-randomized + local search). Due to time and resource constraints during the rebuttal, we did not run extremely large instances, but we additionally report short-time (e.g., 30-second) performance. The results show that heuristic methods are indeed very strong when the MIS size is small, while their performance degrades on larger MIS-sizes (not necessarily larger graphs), where PGD remains competitive.
> > >
> > > We also clarify that our scalability claim primarily refers to the fact that PGD naturally supports GPU parallelization, enabling multiple initializations to be optimized simultaneously via tensor-based gradient computation (please see the GPU results below). While we acknowledge that many GPU-based solvers exist and a comprehensive comparison would be valuable, we believe the current results provide a reasonable empirical motivation for our analysis.
> > >
> > > To provide additional context on practical usefulness of PGD+relax QUBO, prior work such as [NQUBO] demonstrates that gradient-based optimization over relaxed QUBO can be used as a building block to iteratively refine solutions obtained from heuristics by converging to a nearby local minimizer. Our theoretical results can provide support for such practice by showing that, under appropriate conditions, such local minimizers are guaranteed to be binary and feasible.
> > >
> > > **We would like to emphasize that the main contribution of this work is still the theoretical analysis of the optimization landscape, rather than proposing a new solver**.
> > >
> > > We appreciate the reviewers’ feedback and their recognition of the strengths and limitations of our approach.
> > >
> > > 5min (all PGD instances run in python, **rows are ordered according to the MIS size**)
> > > |$(n, p)$|PGD|Gurobi|SLS|GRASP
> > > |-|-|-|-|-|
> > > |(8000,0.7)|10.75|7.75|10.5|11.125||
> > > |(4000,0.5)|16|11.625|16.125|17|
> > > |(2000,0.3)|27|21.875|25.625|27.125|
> > > |(4000,0.3)|28.5|20.5|28.25|29|
> > > |(8000,0.3)|30.875|23.25|30.375|30.625|
> > > |(3000,1/30)|198.5|142.75|194.75|186.125|
> > >
> > > 30sec (all PGD instances are implemented in Python. We do not report Gurobi results under this time limit, as its performance is significantly limited in this regime)
> > > |$(n, p)$|PGD (Parallel 1GPU)|PGD|SLS|GRASP|
> > > |-|-|-|-|-|
> > > |(8000,0.7)|11.125|10.125|10.375|11|
> > > |(4000,0.5)|16.375|15.75|15.75|16.25||
> > > |(2000,0.3)|26.75|25.75|25.625|26.75||
> > > |(4000,0.3)|29|28.125|28.125|28.125||
> > > |(8000,0.3)|31|29.625|29.875|30||
> > > |(3000,1/30)|200.75|191.5|191.75|183.5|
> > >
> > > [DIMES] DIMES: A Differentiable Meta Solver for Combinatorial Optimization Problems, NeurIPS 2022
> > >
> > > [DIFFUSCO] DIFUSCO: Graph-based Diffusion Solvers for Combinatorial Optimization, NeurIPS 2023
> > >
> > > [OptGNN] Are Graph Neural Networks Optimal Approximation Algorithms? NeurIPS 2024
> > >
> > > [CRA] Controlling Continuous Relaxation for Combinatorial Optimization, NeurIPS 2024
> > >
> > > [LwD] Learning What to Defer for Maximum Independent Sets, ICML 2020
> > >
> > > [NQUBO] A Novel Solver for QUBO Problems: Performance Analysis and Comparative Study with State-of-the-Art Algorithms. arXiv 2025

---

### Decision · Program_Chairs · 2026-04-30

**Decision:**

Accept (regular)

**Comment:**

This paper investigates the theoretical properties of local minima in continuous quadratic-penalty relaxations of binary linear programs. The committee reached a clear consensus that the theoretical contributions of this work are strong and significant. Specifically, the authors provide a rigorous framework comprising a diagonal-free structure, integer-valued penalty gradients, and a local repairability condition to guarantee that the local minimizers of the relaxed problem are both binary and feasible. Reviewers appreciated this formal characterization, noting it provides a valuable, principled way to design differentiable proxies for combinatorial optimization problems.

During the review process, the primary concerns raised by the committee centered around the limited empirical baselines, the scalability of the proposed approach, and the numerical stability regarding exponential slack variables. In the rebuttal phase, the authors effectively addressed these concerns by providing additional large-scale benchmarking on the Maximum Independent Set problem against established solvers under a fixed time budget. This new empirical evidence, combined with the authors' clarifications on optimization robustness, sufficiently alleviated the concerns for the majority of the reviewers. Furthermore, one reviewer explicitly championed the paper for acceptance, emphasizing that the theoretical results are convincing, interesting, and highly relevant to the community.

Given the solid theoretical foundation and the authors' constructive rebuttal, the decision is to accept the paper. For the camera-ready version, the authors are expected to broadly incorporate the feedback provided by the reviewers and fulfill the content promises made during the discussion phase. This should include integrating the newly provided large-scale experiments to better support scalability claims, expanding the theoretical discussions regarding complexity and slack expansions, and refining the overall presentation and notation as agreed upon in the author responses.